# GLOBAL ATTENTION IMPROVES GRAPH NETWORKS GENERALIZATION

## ABSTRACT

This paper advocates incorporating a Low-Rank Global Attention (LRGA) module, a computation and memory efficient variant of the dot-product attention (Vaswani et al., 2017), to Graph Neural Networks (GNNs) for improving their generalization power.

To theoretically quantify the generalization properties granted by adding the LRGA module to GNNs, we focus on a specific family of expressive GNNs and show that augmenting it with LRGA provides algorithmic alignment to a powerful graph isomorphism test, namely the 2-Folklore Weisfeiler-Lehman (2-FWL) algorithm. In more detail we: (i) consider the recent Random Graph Neural Network (RGNN) (Sato et al., 2020) framework and prove that it is universal in probability; (ii) show that RGNN augmented with LRGA aligns with 2-FWL update step via polynomial kernels; and (iii) bound the sample complexity of the kernel's feature map when learned with a randomly initialized two-layer MLP.

From a practical point of view, augmenting existing GNN layers with LRGA produces state of the art results in current GNN benchmarks. Lastly, we observe that augmenting various GNN architectures with LRGA often closes the performance gap between different models.

## 1 INTRODUCTION

In many domains, data can be represented as a graph, where entities interact, have meaningful relations and a global structure. The need to be able to infer and gain a better understanding of such data rises in many instances such as social networks, citations and collaborations, chemoinformatics, epidemiology etc. In recent years, along with the major evolution of artificial neural networks, graph learning has also gained a new powerful tool - graph neural networks (GNNs). Since first originated (Gori et al., 2005; Scarselli et al., 2009) as recurrent algorithms, GNNs have become a central interest and the main tool in graph learning.

Perhaps the most commonly used family of GNNs are message-passing neural networks (Gilmer et al., 2017), built by aggregating messages from local neighborhoods at each layer. Since information is only kept at the vertices and propagated via the edges, these models' complexity scales linearly with $|V| + |E|$, where $|V|$ and $|E|$ are the number of vertices and edges in the graph, respectively. In a recent analysis of the expressive power of such models, (Xu et al., 2019a; Morris et al., 2018) have shown that message-passing neural networks are at most as powerful as the first Weisfeiler-Lehman (WL) test, also known as vertex coloring. The $k$-WL tests, are a hierarchy of increasing power and complexity algorithms aimed at solving graph isomorphism. This bound on the expressive power of GNNs led to the design of new architectures (Morris et al., 2018; Maron et al., 2019a) mimicking higher orders of the $k$-WL family, resulting in more powerful, yet complex, models that scale super-linearly in $|V| + |E|$, hindering their usage for larger graphs.

Although expressive power bounds on GNNs exist, empirically in many datasets, GNNs are able to fit the train data well. This indicates that the expressive power of these models might not be the main roadblock to a successful generalization. Therefore, we focus our efforts in this paper on strengthening GNNs from a *generalization* point of view. Towards improving the generalization of GNNs we propose the Low-Rank Global Attention (LRGA) module which can be augmented to any GNN. Standard dot-product global attention modules (Vaswani et al., 2017) apply $|V| \times |V|$

attention matrix to node data with $O(|V|^3)$ computational complexity making them impractical for large graphs. To overcome this barrier, we define a $\kappa$-rank attention matrix, where $\kappa$ is a parameter, that requires $O(\kappa|V|)$ memory and can be applied in $O(\kappa^2|V|)$ computational complexity.

To theoretically justify LRGA we focus on a GNN model family possessing maximal expressiveness (i.e., universal) but vary in the generalization properties of the family members. (Murphy et al., 2019; Loukas, 2019; Dasoulas et al., 2019; Loukas, 2020) showed that adding node identifiers to GNNs improves their expressiveness, often making them universal. In this work, we prove that even adding *random* features to the network's input, as suggested in (Sato et al., 2020), a framework we call Random Graph Neural Network (RGNN), GNN models are universal in probability.

The improved generalization properties of LRGA-augmented GNN models is then showcased for the RGNN framework, where we show that augmenting it with LRGA algorithmically aligns with the 2-folklore WL (FWL) algorithm; 2-FWL is a strictly more powerful graph isomorphism algorithm than vertex coloring (which bounds message passing GNNs). To do so, we adopt the notion of algorithmic alignment introduced in (Xu et al., 2019b), stating that a neural network aligns with some algorithm if it can simulate it with simple modules, resulting in provable improved generalization. We opt to use monimials in the role of simple modules and prove the alignment using polynomial kernels. Lastly, we bound the sample complexity of the model when learning the 2-FWL update rule. Although our bound is exponential in the graph size, it nevertheless implies that RGNN augmented with LRGA can provably learn the 2-FWL step, when training each module independently with two-layer MLP.

We evaluate our model on a set of benchmark datasets including tasks of graph classification and regression, node labeling and link prediction from (Dwivedi et al., 2020; Hu et al., 2020). LRGA improves state of the art performance in most datasets, often with a significant margin. We further perform ablation study in the random features framework to support our theoretical propositions.

## 2 RELATED WORK

**Attention mechanisms.** The first work to use an attention mechanism in deep learning was (Bahdanau et al., 2015) in the context of natural language processing. Ever since, attention has proven to be a powerful module, even becoming the only component in the transformer architecture (Vaswani et al., 2017). Intuitively, attention provides an adaptive importance metric for interactions between pairs of elements, e.g., words in a sentence, pixels in an image or nodes in a graph. A natural drawback of classical attention models is the quadratic complexity generated by computing scores among pairs. Methods to reduce the computation complexity were introduced by (Lee et al., 2018b) which introduced the set-transformer and addressed the problem by inducing point methods used in sparse Gaussian processes. Linearized versions of attention were suggested by (Shen et al., 2020) factorizing the attention matrix and normalizing separate components. Concurrently to the first version of this paper (Anonymous, 2020), Katharopoulos et al. (2020) formulated a linearized attention for sequential data.

**Attention in graph neural networks.** In the field of graph learning, most attention works (Li et al., 2016; Veličković et al., 2018; Abu-El-Haija et al., 2018; Bresson & Laurent, 2017; Lee et al., 2018a) restrict learning the attention scores to the local neighborhoods of the nodes in the graph. Motivated by the fact that local aggregations cannot capture long range relations which may be important when node homophily does not hold, global aggregation in graphs using node embeddings have been suggested by (You et al., 2019; Pei et al., 2020). An alternative approach for going beyond the local neighborhood aggregation utilizes diffusion methods: (Klicpera et al., 2019) use diffusion in a pre-process to replace the adjacency with a sparsified weighted diffusion matrix, while (Zhuang & Ma, 2018) add the diffusion matrix as an additional aggregation operator. LRGA allows *global weighted* aggregations via embedding of the nodes in a low dimension (i.e., rank) space.

**Generalization in graph neural networks.** Although being a pillar stone of modern machine learning, the generalization capabilities of NN are still not very well understood, e.g., see (Bartlett et al., 2017; Golowich et al., 2019). Due to the irregular structure of graph data and the weight sharing nature of GNN, investigating their generalizing capabilities poses an even greater challenge. Despite the nonstandard setting, few works were able to construct generalization bounds for GNN

via *VC dimension* (Scarselli et al., 2018), *uniform stability* (Verma & Zhang, 2019), *Rademacher Complexity* (Garg et al., 2020) and *Neural Tangent Kernel* (Du et al., 2019).

## 3 PRELIMINARIES AND NOTATIONS

We denote a graph by $G = (V, E, \boldsymbol{X})$ where $V$ is the vertex set of size $|V| = n$, $E$ is the edge set, and adjacency $\boldsymbol{A}$. $\boldsymbol{X} = (\boldsymbol{x}_1, \ldots, \boldsymbol{x}_n)^T$ represents the input vertex features. A vertex $v_i \in V$ carries an input feature vector $\boldsymbol{x}_i \in \mathbb{R}^{d_0}$; in turn, $\boldsymbol{X}^l \in \mathbb{R}^{n \times d_l}$ represents the output of the $l^{th}$ layer of a neural network. We denote concatenation along the last dimension with brackets and stacking along a new last dimension with double brackets, i.e., for $\boldsymbol{W}, \boldsymbol{Z} \in \mathbb{R}^{n \times d}$, $[\boldsymbol{W}, \boldsymbol{Z}] \in \mathbb{R}^{n \times 2d}$ and $[\![\boldsymbol{W}, \boldsymbol{Z}]\!] \in \mathbb{R}^{n \times d \times 2}$.

A common form of evaluating GNNs is by their ability to distinguish different graphs, described by *graph isomorphism* which is an equivalence relation between graphs. The isomorphism type tensor of a graph $G$ is a tensor $\mathbf{Y} \in \mathbb{R}^{n^2 \times d_{\mathrm{iso}}}$ which holds the isomorphism types of all pairs $(i, j) \in [n] \times [n]$. Given a pair $(i, j)$, which represents either an edge or a node of graph $G$, $\mathbf{Y}_{i,j}$ summarizes all the information this pair carries in graph $G$. More precisely put, *isomorphism type* is an equivalence relation defined by: $(i, j)$ and $(i', j')$ have the same isomorphism type *iff* the following conditions hold: (i) $i = j \iff i' = j'$; (ii) $\boldsymbol{x}_i = \boldsymbol{x}_{i'}$ and $\boldsymbol{x}_j = \boldsymbol{x}_{j'}$; and (iii) $(i, j) \in E \iff (i', j') \in E$. One way to build an isomorphism type tensor for graph $G$ is $\mathbf{Y} = [\![\boldsymbol{I}, \mathbf{1} \otimes \boldsymbol{X}, \boldsymbol{X} \otimes \mathbf{1}, \boldsymbol{A}]\!]$, where $\boldsymbol{I}$ is the identity matrix, $(\mathbf{1} \otimes \boldsymbol{X})_{i,j,:} = \boldsymbol{x}_j$, and similarly (with a slight abuse of notation) $(\boldsymbol{X} \otimes \mathbf{1})_{i,j,:} = \boldsymbol{x}_i$.

## 4 LOW-RANK GLOBAL ATTENTION (LRGA)

We propose the Low-Rank Global Attention (LRGA) module that can augment any graph neural network layer, denoted here generically as GNN, in the following way:

$$\boldsymbol{X}^{l+1} \leftarrow \left[ \boldsymbol{X}^l, \mathrm{LRGA}(\boldsymbol{X}^l), \mathrm{GNN}(\boldsymbol{X}^l) \right] \tag{1}$$

where the brackets denote concatenation along the feature dimension. The LRGA module is defined for an input feature matrix $\boldsymbol{X} \in \mathbb{R}^{n \times d_{\mathrm{in}}}$ via

$$\mathrm{LRGA}(\boldsymbol{X}) = \left[ \frac{1}{\eta(\boldsymbol{X})} m_1(\boldsymbol{X}) \left( m_2(\boldsymbol{X})^T m_3(\boldsymbol{X}) \right), \; m_4(\boldsymbol{X}) \right] \tag{2}$$

where $m_1, m_2, m_3, m_4 : \mathbb{R}^{n \times d_{\mathrm{in}}} \to \mathbb{R}^{n \times \kappa}$ are MLPs operating on the feature dimension, that is $m(\boldsymbol{X}) = [m(\boldsymbol{x}_1), \ldots, m(\boldsymbol{x}_n)]^T$, and $\kappa \in \mathbb{N}_0$ is a parameter representing the *rank* of the attention module. Lastly, $\eta$ is a normalization factor:

$$\eta(\boldsymbol{X}) = \frac{1}{n} \left( \mathbf{1}^T m_1(\boldsymbol{X}) \right) \left( m_2(\boldsymbol{X})^T \mathbf{1} \right), \tag{3}$$

where $\mathbf{1} = (1, 1, \ldots, 1)^T \in \mathbb{R}^n$. The matrix $\eta(\boldsymbol{X})^{-1} m_1(\boldsymbol{X}) m_2(\boldsymbol{X})^T$ can be thought of as a $\kappa$-rank attention matrix that acts globally on the graph's node features.

**Computational complexity.** Standard attention models (Vaswani et al., 2017; Luong et al., 2015) require explicitly computing the attention score between all possible pairs in the set, meaning that its memory requirement and computational cost scales as $O(n^2)$. This makes global-attention seem impractical for large sets, or large graphs in our case. We address the global attention computational challenge by working with bounded rank (i.e., $\kappa$) attention matrices, and avoid the need to construct the attention matrix in memory by replacing the standard entry-wise normalization (softmax or tanh) with a the global normalization $\eta$. In turn, the memory requirement of LRGA is $O(n\kappa)$, and using low rank matrix-vector multiplications LRGA allows applying global attention in $O(n\kappa^2)$ computation cost.

**Permutation Equivariance.** A common demand from GNN architectures is to respect the graph representation symmetries, namely the ordering of nodes (Maron et al., 2019b). As shown in (Lee et al., 2018b) the set attention module is permutation equivariant. The same matrix product structure of the LRGA makes this module also permutation equivariant.

## 5 THEORETICAL ANALYSIS

In this section we establish the theoretical underpinning for LRGA. Since we want to analyse the generalization power added by LRGA, we focus on a family of GNNs with unbounded expressive power *in probability* (RGNN). Under this model we show the benefit of augmenting GNNs with LRGA in terms of improved generalization via the notion of *algorithmic alignment* with a powerful graph isomorphism testing algorithm (2-FWL).

### 5.1 RANDOM GRAPH NEURAL NETWORKS

We analyse LRGA under the framework of Random Graph Neural Networks (RGNNs):

**Definition 1** (Random Graph Neural Network). *Let $\mathcal{D}$ be a probability distribution of zero mean and variance $c$, and $G = (V, E, \boldsymbol{X})$ a graph. RGNN is a GNN variant with random input features sampled at every forward pass i.e., the input to the network is $[\boldsymbol{X}, \boldsymbol{R}]$ where $\boldsymbol{R}$ are i.i.d. samples $\boldsymbol{R} \in \mathbb{R}^{n \times d} \sim \mathcal{D}$.*

RGNN, suggested by Sato et al. (2020), has related variants (Loukas, 2020; 2019; Murphy et al., 2019) that use node identifiers or distinctive features, which can be viewed as *constant* random features, in order to break symmetry between isomorphic nodes. Such models are proven to be universal but lose their inherent equivariance due to arbitrary prescription of node identifiers. We choose to work in the seemingly more limited setting of RGNN, which allows the network to distinguish between different nodes but does not overfit specific identifiers. Our main claims regarding this framework is that RGNN is both universal in probability and equivariant in expectation.

**Proposition 1** (Universal). *RGNN can approximate an arbitrary continuous graph function given random features sampled from a bounded distribution $\mathcal{D}$.*

Here approximation is in a probabilitic sense: Let $\Omega \subset \mathbb{R}^{n \times d_0} \times \mathbb{R}^{n^2}$ be a compact set of graphs, $[\boldsymbol{X}, \boldsymbol{A}] \in \Omega$, where $\boldsymbol{A} \in \mathbb{R}^{n^2}$ is the adjacency matrix. Then, given a continuous graph function $f$ defined over $\Omega$ and arbitrary $\varepsilon, \delta > 0$, there exist network parameters and $d$ so that $P(|\mathrm{GNN}([\boldsymbol{X}, \boldsymbol{R}]) - f([\boldsymbol{X}, \boldsymbol{A}])| < \varepsilon) > 1 - \delta$, for all graphs $[\boldsymbol{X}, \boldsymbol{A}] \in \Omega$. Proposition 1 holds for GNN variants with a global attribute block such as (Battaglia et al., 2018). The proof is based on the idea that random features allow the GNN to transfer the graph's connectivity information to the node features. Once all graph information is encapsulated at the nodes, we exploit the universality of set functions (Zaheer et al., 2017) to get universality. The full proof is in Appendix A. To the best of our knowledge this is the first result proving universality under the random feature assumption.

**Proposition 2** (Equivariant in expectation). *RGNN is permutation equivariant in expectation.*

Changing the random features at each forward pass allows RGNN to preserve equivariance in expectation. Indeed, equivariance of GNN implies that $\mathrm{GNN}(\boldsymbol{P} \cdot [\boldsymbol{X}, \boldsymbol{R}]) = \boldsymbol{P} \cdot \mathrm{GNN}([\boldsymbol{X}, \boldsymbol{R}])$, for any permutation matrix $\boldsymbol{P}$ and input $[\boldsymbol{X}, \boldsymbol{R}]$. Taking the expectation of both sides w.r.t. $\boldsymbol{R} \sim \mathcal{D}$, noting that $\boldsymbol{P}\boldsymbol{R} \sim \boldsymbol{R}$ and using linearity of expectation we get equivariance in expectation.

### 5.2 RGNN AUGMENTED WITH LRGA ALIGNS WITH 2-FWL

In this section we will formulate our main theoretical result, Theorem 1, stating that augmenting RGNN with LRGA algorithmically aligns with a powerful graph isomorphism testing algorithm called 2-Folklore Weisfeiler-Lehman (2-FWL) (Grohe & Otto, 2015; Grohe, 2017). We will first introduce the notion of algorithmic alignment and the 2-FWL algorithm, then formulate our main theorem, and continue in the next section with a proof.

**Algorithmic alignment.** The notion of *algorithmic alignment* was introduced in Xu et al. (2019b) as a framework for exploring effective neural architectures for certain tasks. A neural network $\mathcal{N}$ is said to be aligned with an algorithm $\mathcal{A}$ if $\mathcal{N}$ can simulate $\mathcal{A}$ by a composition of modules, and each module is "simple", or learnable, i.e., have bounded (hopefully low) sample complexity. For example, message passing networks can simulate the vertex coloring algorithm (Xu et al., 2019a; Morris et al., 2018) and therefore message passing can be seen as algorithmically aligned with vertex coloring. Intuitively, algorithmic alignment introduces an inductive bias that improves the sample complexity. Our definition of algorithmic alignment is a slightly stricter version:

**Definition 2** (Monomial Algorithmic Alignment). *A neural network $\mathcal{N}$ aligns with algorithm $\mathcal{A}$ if $\mathcal{N}$ can simulate $\mathcal{A}$ by learning only monomial functions, i.e., $f(\boldsymbol{x}) = \boldsymbol{x}^{\boldsymbol{\alpha}}$, where $\boldsymbol{x} \in \mathbb{R}^d$, $\boldsymbol{\alpha} \in \mathbb{N}^d$, and $\boldsymbol{x}^{\boldsymbol{\alpha}} = x_1^{\alpha_1} \cdot \ldots \cdot x_d^{\alpha_d}$.*

To motivate this choice of monomials as "simple" functions we note that (Arora et al., 2019; Xu et al., 2019b) show a sample complexity bound for even-power polynomials learned by (two-layer) MLPs and we extend it to general monomials in the following proposition proved in Appendix E:

**Proposition 3.** *Let a two layer MLP trained with gradient descent be denoted as the learning algorithm $\mathcal{A}'$. The monomial $g(\boldsymbol{x}) = \boldsymbol{x}^{\boldsymbol{\alpha}}$, $\boldsymbol{x} \in \mathbb{R}^d$, of degree $n$, $|\boldsymbol{\alpha}| \leq n$, is PAC learnable with $\mathcal{A}'$ with a sample complexity bound:*

$$\mathcal{C}_{\mathcal{A}'}(g, \epsilon, \delta) = \mathcal{O}\left(\frac{C_{n,d} + \log(1/\delta)}{\epsilon^2}\right),$$

$C_{n,d} = \left(n^2 + 1\right)^{(n+1)/2} c_{n,d}$, $\varepsilon > 0$ *is the error parameter and* $\delta \in (0, 1)$ *the failure probability.*

The asymptotic behaviour of $c_{n,d}$ is out of the scope of this paper. Therefore, a monomial algorithmic alignment of $\mathcal{N}$ to $\mathcal{A}$ means (under the assumptions and sequential training method of Theorem 3.6 in Xu et al. (2019b)) that $\mathcal{A}$ is learnable by $\mathcal{N}$.

**2-Folklore Weisfeiler-Lehman (2-FWL) Algorithm.** 2-FWL is part of the $k$-WL hierarchy of polynomial-time (approximate) graph isomorphism iterative algorithms that recolor $k$-tuples of vertices at each step according to neighborhoods aggregation. Upon reaching a stable coloring, the algorithm terminates and if the histograms of colors of two graphs are not the same then the graphs are deemed not isomorphic. The 2-FWL algorithm is equivalent to 3-WL, strictly stronger than vertex coloring (2-WL) which bounds the expressive power of GNNs.

In more detail, let $\mathbf{Y}^0 \in \mathbb{R}^{n^2 \times d_{\text{iso}}}$ represent the isomorphism types of a given graph $G = (V, E, \boldsymbol{X})$, that is $\mathbf{Y}^0_{i,j} \in \mathbb{R}^{d_{\text{iso}}}$ represents the isomorphism type of the pair $(i, j)$. The 2-FWL algorithm is initialized with $\mathbf{Y}^0$. Let $\mathbf{Y}^l \in \mathbb{R}^{n^2 \times d_l}$ denote the coloring tensor after the $l^{th}$ update step. An update step in the algorithm aggregates information from the multiset of neighborhood colors for each pair. We represent the multiset of neighborhood colors of the tuple $(i, j)$ 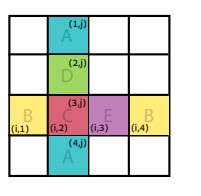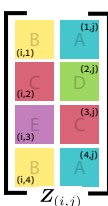

with a matrix $\boldsymbol{Z}^l_{(i,j)} \in \mathbb{R}^{n \times 2d_l}$. That is, any permutation of the rows of $\boldsymbol{Z}^l_{(i,j)}$ represent the same multiset. The rows of $\boldsymbol{Z}^l_{(i,j)}$, which represent the elements in the multiset, are $\boldsymbol{z}_k = [\mathbf{Y}^l_{i,k}, \mathbf{Y}^l_{k,j}] \in \mathbb{R}^{2d_l}$, $k \in [n]$. See the inset for an illustration. The 2-FWL update step of a pair $(i, j)$ from $\mathbf{Y}^l$ to $\mathbf{Y}^{l+1}$ concatenates the previous pair's color and an encoding of the multiset of neighborhoods colors:

$$\mathbf{Y}^{l+1}_{i,j} = \left[\mathbf{Y}^l_{i,j}, \text{ENC}\left(\boldsymbol{Z}^l_{(i,j)}\right)\right] \tag{4}$$

where $\text{ENC} : \mathbb{R}^{n \times 2d_l} \to \mathbb{R}^{d_{\text{enc}}}$ is a multiset injective map invariant to the row-order of its input.

**Main result.** Consider the 2-FWL update rule in equation 4 and let $\boldsymbol{Y}^{l+1} \in \mathbb{R}^{n^2}$ denote (arbitrary) single feature dimension pealed off $\mathbf{Y}^{l+1} \in \mathbb{R}^{n^2 \times d_{l+1}}$; we call $\boldsymbol{Y}^{l+1}$ a single-head of the update rule. Then,

**Theorem 1.** *LRGA augmented RGNN algorithmically aligns with a single head 2-FWL update step.*

A corollary of this theorem is:

**Corollary 1.** *Multi-head LRGA augmented RGNN algorithmically aligns with 2-FWL.*

Multi-head LRGA is a module of the form $[\boldsymbol{X}^l, \text{LRGA}_1(\boldsymbol{X}^l), \ldots, \text{LRGA}_k(\boldsymbol{X}^l), \text{GNN}(\boldsymbol{X}^l)]$, which is an equivalent to multi-head self-attention. In practice, we found single-head LRGA to be on par performance-wise with multi-head LRGA and therefore we focus on the single-head version in the experimental section.

### 5.3 PROOF OF THEOREM 1

To prove Theorem 1 we need to show RGNN augmented with LRGA can simulate one head of the 2-FWL update step using only monomials as learnable functions. We achieve that by the following

steps: (i) introduce the notion of node factorization to encode $n \times n$ tensor data as node features; (ii) show that RGNN can approximate node factorization of the graph's isomorphism type tensor with a single GNN layer using learnable monomial functions; (iii) show that 2-FWL update step can be formulated using matrix multiplication of monomial functions; and (iv) show LRGA can approximate a single head 2-FWL update step using learnable monomials.

**Part (i).** We start with the definition of node feature factorization:

**Definition 3** (Node factorization). *Let* $\mathbf{Y} \in \mathbb{R}^{n^2 \times d}$ *be a tensor.* $\mathbf{X} \in \mathbb{R}^{n \times D}$ *is called node factorization of* $\mathbf{Y}$ *if there exists a block structure* $\mathbf{X} = \begin{bmatrix} \mathbf{X}^1, \ldots, \mathbf{X}^k \end{bmatrix}$ *so that* $\mathbf{Y} = \begin{bmatrix} \mathbf{X}^{s_1}(\mathbf{X}^{t_1})^T, \ldots, \mathbf{X}^{s_d}(\mathbf{X}^{t_d})^T \end{bmatrix}$, *where* $(s_1, t_1), \ldots, (s_d, t_d) \in [k] \times [k]$ *are index pairs.*

Note that for all $i, j \in [n]$ we have $\mathbf{Y}_{i,j} = \begin{bmatrix} \langle \boldsymbol{x}_i^{s_1}, \boldsymbol{x}_j^{t_1} \rangle, \ldots, \langle \boldsymbol{x}_i^{s_d}, \boldsymbol{x}_j^{t_d} \rangle \end{bmatrix} \in \mathbb{R}^d$. Lets illustrate the definition with an example. Let $\boldsymbol{A} \in \{0, 1\}^{n \times n}$ be the adjacency matrix of some graph $G$, and for simplicity assume that there are no node features. Then, the isomorphism type tensor of $G$ is $\mathbf{Y}^0 = \llbracket \boldsymbol{I}, \boldsymbol{A} \rrbracket \in \mathbb{R}^{n^2 \times 2}$. One possible way of node factoring $\mathbf{Y}^0$ is using the SVD decomposition of the adjacency matrix $\boldsymbol{A}$. Note that node factorization is not unique.

**Part (ii).**

**Proposition 4.** *RGNN with skip connection can approximate node factorization of the isomorphism type tensor* $\mathbf{Y}^0$.

*Proof.* We will prove the case of graph $G = (V, E)$, i.e., with no vertex features; the general case can be found in Appendix D. Let $\boldsymbol{R} \in \mathbb{R}^{n \times d}$ be a random node features matrix sampled i.i.d. from $\mathcal{D}$. A single layer of standard message passing can represent $\text{GNN}(\boldsymbol{R}) = d^{-0.5}[\boldsymbol{A}\boldsymbol{R}, \boldsymbol{R}]$, which requires learning only first degree (linear) monomials in the GNN's learnable parts. Furthermore, $\text{GNN}(\boldsymbol{R})$ is an approximate node factorization of $\mathbf{Y}^0$, since $d^{-1} \llbracket \boldsymbol{R}\boldsymbol{R}^T, \boldsymbol{A}\boldsymbol{R}\boldsymbol{R}^T \rrbracket \approx \llbracket \boldsymbol{I}, \boldsymbol{A} \rrbracket = \mathbf{Y}^0$, where the approximation error $d^{-1}\boldsymbol{R}\boldsymbol{R}^T \approx \boldsymbol{I}$ can be bounded using the result in Appendix A. $\square$

**Part (iii).** As shown in (Maron et al., 2019a) the encoding function ENC from the 2-FWL update rule (see equation 4) can be expressed as follows (derivation can be found in Appendix B):

$$\mathbf{Y}^{l+1} = \left[\!\!\left[ \mathbf{Y}, \left[ \mathbf{Y}^{\boldsymbol{\beta}} \mathbf{Y}^{\boldsymbol{\gamma}} \mid (\boldsymbol{\beta}, \boldsymbol{\gamma}) \in \mathbb{N}_0^{2d}, |\boldsymbol{\beta}| + |\boldsymbol{\gamma}| \le n \right] \right]\!\!\right] \tag{5}$$

where for notational simplicity we denote $\mathbf{Y} = \mathbf{Y}^l$ and $d = d_l$. By $\mathbf{Y}^{\boldsymbol{\beta}}$ we mean that we apply the multi-power $\boldsymbol{\beta}$ to the feature dimension, i.e., $(\mathbf{Y}^{\boldsymbol{\beta}})_{i,j} = \mathbf{Y}_{i,j}^{\boldsymbol{\beta}}$. Therefore, computing the multisets encoding amounts to calculating monomials $\mathbf{Y}^{\boldsymbol{\beta}}, \mathbf{Y}^{\boldsymbol{\gamma}}$ and their matrix multiplications $\mathbf{Y}^{\boldsymbol{\beta}} \mathbf{Y}^{\boldsymbol{\gamma}}$.

**Part (iv).**

**Proposition 5.** *The node factorization of each head of* $\mathbf{Y}^{l+1}$, *the result of* 2*-FWL update step, can be approximated via LRGA module applied to node factorization of* $\mathbf{Y} = \mathbf{Y}^l$. *The MLPs in the LRGA approximation need to learn only monomial functions.*

*Proof.* Let $\boldsymbol{X} = [\boldsymbol{X}^1, \ldots, \boldsymbol{X}^k] \in \mathbb{R}^{n \times D}$ be a node factorization of $\mathbf{Y} = \mathbf{Y}^l$. The 2-FWL update step requires computation of polynomials of the form $\mathbf{Y}^{\boldsymbol{\beta}}$ as shown in equation 5. Using the node factorization of $\mathbf{Y}$, $\mathbf{Y}_{i,j} = \begin{bmatrix} \langle \boldsymbol{x}_i^{s_1}, \boldsymbol{x}_j^{t_1} \rangle, \ldots, \langle \boldsymbol{x}_i^{s_d}, \boldsymbol{x}_j^{t_d} \rangle \end{bmatrix} \in \mathbb{R}^d$, we can write:

$$\mathbf{Y}_{i,j}^{\boldsymbol{\beta}} = \prod_{l=1}^{d} \langle \boldsymbol{x}_i^{s_l}, \boldsymbol{x}_j^{t_l} \rangle^{\beta_l} = \prod_{l=1}^{d} \langle \varphi_{\beta_l}(\boldsymbol{x}_i^{s_l}), \varphi_{\beta_l}(\boldsymbol{x}_j^{t_l}) \rangle = \prod_{l=1}^{d} \langle \varphi_{\beta_l}(\boldsymbol{x}_i^{s}), \varphi_{\beta_l}(\boldsymbol{x}_j^{t}) \rangle$$
$$= \langle \varphi_{\boldsymbol{\beta}}(\boldsymbol{x}_i^{s}), \varphi_{\boldsymbol{\beta}}(\boldsymbol{x}_j^{t}) \rangle \tag{6}$$

where the second equality is using the feature maps $\varphi_{\beta_l}$ of the (homogeneous) polynomial kernels (Vapnik, 1998), $\langle \boldsymbol{x}_1, \boldsymbol{x}_2 \rangle^{\beta_l}$; the third equality is reformulating the feature maps $\varphi_{\beta_l}$ on the vectors $\boldsymbol{x}_i^s = [\boldsymbol{x}_i^{s_1}, \ldots, \boldsymbol{x}_i^{s_d}]$, and $\boldsymbol{x}_i^t = [\boldsymbol{x}_i^{t_1}, \ldots, \boldsymbol{x}_i^{t_d}]$; and the last equality is due to the closure of kernels to multiplication. We denote the final feature map by $\varphi_{\boldsymbol{\beta}}$.

Now, let $\psi_{\boldsymbol{\beta}}(\boldsymbol{x}_i) = \varphi_{\boldsymbol{\beta}}(\boldsymbol{x}_i^s)$ and $\phi_{\boldsymbol{\beta}}(\boldsymbol{x}_i) = \varphi_{\boldsymbol{\beta}}(\boldsymbol{x}_i^t)$ then we have:

$$\mathbf{Y}^{\boldsymbol{\beta}} = \psi_{\boldsymbol{\beta}}(\boldsymbol{X}) \phi_{\boldsymbol{\beta}}(\boldsymbol{X})^T,$$

where $\psi_{\boldsymbol{\beta}}(\boldsymbol{X})$ is applying $\psi_{\boldsymbol{\beta}}$ to every row of $\boldsymbol{X}$. Therefore, arbitrary head of $\mathbf{Y}^{l+1}$, i.e., of the form $\mathbf{Y}^{\boldsymbol{\beta}}\mathbf{Y}^{\boldsymbol{\gamma}}$, can be written directly as a function of $\boldsymbol{X}$ using the feature maps $\phi_{\boldsymbol{\beta}}, \psi_{\boldsymbol{\beta}}, \phi_{\boldsymbol{\gamma}}, \psi_{\boldsymbol{\gamma}}$:

$$\mathbf{Y}^{\boldsymbol{\beta}}\mathbf{Y}^{\boldsymbol{\gamma}} = \psi_{\boldsymbol{\beta}}(\boldsymbol{X})\phi_{\boldsymbol{\beta}}(\boldsymbol{X})^T \psi_{\boldsymbol{\gamma}}(\boldsymbol{X})\phi_{\boldsymbol{\gamma}}(\boldsymbol{X})^T. \tag{7}$$

A node factorization of the head $\mathbf{Y}^{\boldsymbol{\beta}}\mathbf{Y}^{\boldsymbol{\alpha}}$ is therefore $\left[\psi_{\boldsymbol{\beta}}(\boldsymbol{X})\phi_{\boldsymbol{\beta}}(\boldsymbol{X})^T \psi_{\boldsymbol{\gamma}}(\boldsymbol{X}), \phi_{\boldsymbol{\gamma}}(\boldsymbol{X})\right]$. Recalling the structure of the LRGA module introduced in equation 2: $\text{LRGA}(\boldsymbol{X}) = \left[\eta(\boldsymbol{X})^{-1}m_1(\boldsymbol{X})\left(m_2(\boldsymbol{X})^T m_3(\boldsymbol{X})\right), m_4(\boldsymbol{X})\right]$, to implement the 2-FWL head the MLPs $m_1, m_2, m_3, m_4$ need to learn the polynomial feature maps formulated in equation 7: $m_1 \approx \psi_{\boldsymbol{\beta}}$, $m_2 \approx \phi_{\boldsymbol{\beta}}$, $m_3 \approx \psi_{\boldsymbol{\gamma}}$, and $m_4 \approx \phi_{\boldsymbol{\gamma}}$. Every coordinate of these feature maps is a monomial (proof of this fact in Appendix C). Lastly, note that 2-FWL tensors $\mathbf{Y}^l$ are insensitive to global scaling and therefore the normalization $\eta$ has no theoretical influence (it is assumed non-zero). $\square$

## 6 EXPERIMENTS

We evaluated our method on various tasks including graph regression, graph classification, node classification and link prediction. The datasets we used are from two benchmarks: (i) benchmarking GNNs (Dwivedi et al., 2020); and (ii) Open Graph Benchmark (OGB) (Hu et al., 2020). Each benchmark has its own evaluation protocol designed for a fair comparison among different models. These protocols define consistent splits of the data to train/val/test sets, set a budget on the size of the models (OGB), define a stopping criterion for reporting test results and require training with several different initializations to measure the stability of the results. We followed these protocols.

**Baselines.** We compare performance with the following state of the art baselines: *GCN* (Kipf & Welling, 2016), *GraphSAGE* (Hamilton et al., 2017), *GIN* (Xu et al., 2019a), *GAT* (Veličković et al., 2018), *GatedGCN* (Bresson & Laurent, 2017), *Node2Vec* (Grover & Leskovec, 2016), DeepWalk (Perozzi et al., 2014) and *MATRIX FACTORIZATION* (Hu et al., 2020).

**Attention Ablation.** We compared the performance of different versions of global attention modules. The experiment was conduced on the ZINC dataset and compared performance on the GCN, GAT and GatedGCN models.

**Random Features Evaluation.** In addition, we also conducted a set of experiments with the random feature framework. In this experiment we focused on the PATTERN node classification dataset from (Dwivedi et al., 2020) and evaluated a variety of models under the RGNN framework.

**Rank Ablation Study.** In this experiment we examined the relation between the rank parameter $\kappa$, which can limit the expressiveness of the attention module, and the network performance. Results are presented in Appendix G.

**Implementation details of LRGA.** We implemented the LRGA module according to the description in Section 4 (equations 2, 3) using the pytorch framework and the DGL (Wang et al., 2019) and Pytorch geometric (Fey & Lenssen, 2019) libraries. Each LRGA module contains 4 MLPs $m_1, m_2, m_3, m_4$. Each $m_i : \mathbb{R}^d \rightarrow \mathbb{R}^{\kappa}$ is a single layer MLP (linear with ReLU activation). The implementation of a layer is according to equation 2, where in practice we added another single layer MLP, $m_5 : \mathbb{R}^{d+2\kappa+d_{GNN}} \rightarrow \mathbb{R}^d$, for the purpose of reducing the feature dimension size. In the OGB benchmark dataset we did not use the skip connections (better performance), and as advised in (Wang et al., 2019), we used batch and graph normalization at each layer.

### 6.1 BENCHMARKING GRAPH NEURAL NETWORKS (DWIVEDI ET AL., 2020)

**Datasets.** This benchmark contains 6 main datasets (full description in appendix H.1) : (i) ZINC, graph regression task of molecular dataset evaluated with MAE metric; (ii) MNIST and CIFAR10, the image classification problem converted to graph classification using a super-pixel representation (Knyazev et al., 2019); (iii) CLUSTER and PATTERN, node classification tasks which aim to classify embedded node structures (Abbe, 2017); (iv) TSP, a link prediction variation of the Traveling Salesman Problem (Joshi et al., 2019) on 2D Euclidean graph.

**Evaluation protocol.** All models were evaluated with two different sets of parameter budgets and restrictions. The first set restricted to have roughly $100K$ parameters and 4 layers, while the second set of experiments has a budget of roughly $500K$ parameters and up to 16 layers. The learning rate and its decay are set according to a predetermined scheduler using the validation loss. The stopping criterion is set to when the learning rate reaches a specified threshold. All results are averaged over a set of predetermined fixed seeds and standard deviation is reported as well.

Table 1: Performance on the benchmarking GNN datasets. In bold: better performance between LRGA augmented and vanilla models; note the parameter (#) budget. Blue represents best performance with the 100K budget and red with the 500K budget.

| Model | PATTERN # | Acc ± std | CLUSTER # | Acc ± std | ZINC # | MAE ± std | MNIST # | Acc ± std | CIFAR10 # | Acc ± std | TSP # | F1 ± std |
|---|---|---|---|---|---|---|---|---|---|---|---|---|
| GCN | 100K | 63.88 ± 0.07 | 101K | 53.44 ± 2.02 | 103K | 0.459 ± 0.006 | 101K | 90.70 ± 0.21 | 101K | 55.71 ± 0.38 | 95K | 0.630 ± 0.001 |
| LRGA + GCN | 90K | **83.09 ± 0.73** | 91K | **68.44 ± 0.16** | 92K | **0.448 ± 0.009** | 91K | **97.63 ± 0.11** | 91K | **65.80 ± 0.43** | 97K | **0.702 ± 0.001** |
| GAT | 109K | 75.82 ± 1.82 | 110K | 57.73 ± 0.32 | 102K | 0.475 ± 0.007 | 110K | 95.53 ± 0.20 | 110K | 64.22 ± 0.45 | 96K | 0.671 ± 0.002 |
| LRGA + GAT | 90K | **82.54 ± 0.71** | 91K | **69.05 ± 0.05** | 92K | **0.421 ± 0.020** | 90K | **97.47 ± 0.16** | 90K | **68.00 ± 0.13** | 97K | **0.680 ± 0.003** |
| GatedGCN | 104K | 84.48 ± 0.12 | 104K | 60.40 ± 0.41 | 105K | 0.375 ± 0.003 | 104K | 97.34 ± 0.14 | 104K | 67.31 ± 0.31 | 97K | **0.808 ± 0.003** |
| LRGA + GatedGCN | 93K | **85.09 ± 0.11** | 93K | **69.28 ± 0.16** | 94K | **0.355 ± 0.010** | 93K | **98.20 ± 0.03** | 93K | **70.65 ± 0.18** | 97K | 0.807 ± 0.001 |
| GCN | 500K | 71.89 ± 0.33 | 501K | 68.49 ± 0.97 | 505K | **0.367 ± 0.011** | 504K | 91.39 ± 0.25 | 504K | 54.84 ± 0.44 | - | - |
| LRGA + GCN | 400K | **84.55 ± 0.57** | 400K | **76.01 ± 0.67** | 501K | 0.377 ± 0.009 | 463K | **98.34 ± 0.06** | 463K | **68.27 ± 0.46** | - | |
| GAT | 526K | 78.27 ± 0.18 | 528K | 70.58 ± 0.44 | 531K | 0.384 ± 0.007 | 441K | 96.50 ± 0.18 | 442K | 66.11 ± 0.98 | - | - |
| LRGA + GAT | 533K | **85.82 ± 0.42** | 267K | **76.16 ± 0.34** | 536K | **0.360 ± 0.004** | 476K | **98.41 ± 0.08** | 476K | **71.57 ± 0.26** | - | - |
| GatedGCN | 502K | 85.56 ± 0.01 | 502K | 73.84 ± 0.32 | 504K | 0.282 ± 0.015 | 500K | 98.24 ± 0.04 | 500K | 71.33 ± 0.39 | - | - |
| LRGA + GatedGCN | 486K | 85.81 ± 0.31 | 438K | **76.39 ± 0.13** | 446K | **0.249 ± 0.011** | 486K | **98.47 ± 0.16** | 487K | **73.48 ± 0.29** | - | - |

**Results.** Table 1 summarizes the results of training and evaluating our model according to the evaluation protocol; We observe that LRGA improves GNN performance, often by a large margin, across all models and datasets, besides GCN on ZINC and GatedGCN in TSP, supporting our claim for improved generalization. We further note that SOTA in all datasets except TSP is achieved with LRGA augmented GNNs. In some datasets, such as CLUSTER and PATTERN, LRGA reaches top and roughly equivalent performance for all models it augmented, which emphasizes the empirical contribution of LRGA independently of the GNN variant.

## 6.2 LINK PREDICTION DATASETS FROM THE OGB BENCHMARK (HU ET AL., 2020)

**Datasets.** We further evaluate LRGA on semi-supervised learning tasks including graphs with hundreds of thousands of nodes, from the OGB benchmark: (i) ogbl-ppa, a graph of proteins and biological connections as edges ;(ii) ogbl-collab, an authors collaborations graph; (iii) ogbl-ddi drug interaction network. The evaluation metric for all of the tasks is Hits@K; more details in appendix H.2.

Table 2: Performance on the link prediction tasks from the OGB benchmark

| Model | ogbl-ppa # Param | Hits@100±std | ogbl-collab # Param | Hits@50±std | ogbl-ddi # Param | Hits@20±std |
|---|---|---|---|---|---|---|
| Node2vec | 7.3M | 0.223 ± 0.008 | 30M | 0.489 ± 0.005 | 645K | 0.233 ± 0.021 |
| DeepWalk | 150M | 0.289 ± 0.015 | 61M | 0.504 ± 0.003 | 11M | 0.264 ± 0.061 |
| MF | 147M | 0.323 ± 0.009 | 60M | 0.389 ± 0.003 | 1.2M | 0.137 ± 0.047 |
| GraphSage | 424K | 0.165 ± 0.024 | 460K | 0.481 ± 0.008 | 1.4M | 0.539 ± 0.047 |
| GCN | 278K | 0.187 ± 0.013 | 296K | 0.447 ± 0.011 | 1.2M | 0.370 ± 0.050 |
| LRGA + GCN | 814K | **0.342 ± 0.016** | 1M | **0.522 ± 0.007** | 1.5M | **0.623 ± 0.091** |

**Evaluation protocol.** All models have a hidden layer of size 256 and the number of layers is 3 in ogbl-ppa and ogbl-collab and 2 in ogbl-ddi. Test results are reported by the best validation epoch averaged over 10 random seeds.

**Results.** Table 2 summarizes the results on the link prediction tasks. It should be noted that the first three rows correspond to node embedding methods where the rest are GNNs. Augmenting GCN with LRGA achieves SOTA results on those datasets, while still using order of magnitude less parameters than the node embedding runner-up method.

## 6.3 ATTENTION ABLATION

Table 3: Attention ablation table. Various GNNs augmented with attention variants on the ZINC dataset. Bold represent best performance and blue represent second best.

| Model | LRGA MAE ± std | LRGA no $m_4$ MAE ± std | Polynomial kernel (degree 2) MAE ± std | Polynomial kernel (degree 4) MAE ± std | Exponential kernel MAE ± std | RBF kernel MAE ± std |
|---|---|---|---|---|---|---|
| GCN | 0.448 ± 0.009 | **0.440 ± 0.006** | 0.464 ± 0.011 | 0.467 ± 0.008 | 0.450 ± 0.011 | 0.457 ± 0.007 |
| GAT | **0.421 ± 0.020** | 0.435 ± 0.026 | 0.460 ± 0.011 | 0.475 ± 0.014 | 0.439 ± 0.016 | 0.452 ± 0.003 |
| GatedGCN | 0.355 ± 0.010 | 0.363 ± 0.008 | 0.363 ± 0.008 | 0.370 ± 0.011 | **0.351 ± 0.028** | 0.371 ± 0.005 |

The LRGA model (equation 2) applies the low-rank attention matrix $S = \eta(X)^{-1}m_1(X)m_2(X)^T$ to the node features $m_3(X)$, that, together with $m_4(X)$, align with node factorization of 2-FWL head. In this experiment we have tested two variations of LRGA: First, removing the $m_4$ component; and second, replacing $S$ with standard, kernel-based attention matrices (Tsai et al., 2019). Results of

incorporating the different attention mechanisms to GCN, GAT, and GatedGCN and experimenting with the ZINC dataset are summarized in Table 3. First, it seems incorporating $m_4$ explicitly in the LRGA module compares mostly favorably to LRGA model with no $m_4$. We attribute that mainly to the algorithmic alignment of the full LRGA model with 2-FWL, and in particular to the encoding of 2-FWL neighborhood multisets. Second, as indicated in (Tsai et al., 2019), the attention matrix could be expressed using a kernel function, $\mathbf{S}_{i,j} = (\sum_{\ell=1}^{n} k(\mathbf{x}_i, \mathbf{x}_\ell))^{-1} k(\mathbf{x}_i, \mathbf{x}_j)$. We replace the low-rank attention matrix $\mathbf{S}$ in the LRGA module with attention matrices defined via different kernels $k$: a polynomial kernel (of degree 2 and 4); exponential kernel (which is equivalent to the classical self-attention (Vaswani et al., 2017)) and radial basis function (RBF) kernel. A full definition of the different kernels is provided in Appendix F. Note that the proof of Theorem 1 utilizes a kernel defined by a polynomials feature map to align with the 2-FWL head. As the table shows, with the expection of the exponential kernel on GatedGCN, LRGA achieve superior result across all the models. The major advantage of LRGA over the other kernels in that it does not require to explicitly compute and store in memory the attention matrix, and exploit the low rank structure for fast multiplication.

## 6.4 RANDOM FEATURES ABLATION

In this experiment we wanted to validate the theoretical analysis presented at section 5. The dataset for this evaluation is the PATTERN dataset, which is originally equipped with random features, but in contrast to the RGNN framework those features are sampled only once at the dataset creation stage. We evaluated the different models according to the RGNN framework, i.e., resample the features with every forward pass. The features were sampled from a zero mean Gaussian distribution with variance $\frac{1}{d}$, where $d$ is the input feature dimension. The evaluation protocol is the same as the one used in section 6.1 and we followed the $500K$ budget. As seen from table 4, using alternating random features improves performance for all the models. *GIN* and *GraphSage* do not appear in the main table but according to (Dwivedi et al., 2020) achieves $85.39\%$ and $50.49\%$ respectively. The LRGA augmented RGNN models maintain their superiority (even presenting a small improvement compared to Table 1) and serve as an empirical validation to our main theorem.

Table 4: Random Features Evaluation

| Model | PATTERN |
|---|---|
| | Acc $\pm$ std |
| GCN | 74.891 $\pm$ 0.713 |
| LRGA + GCN | 84.118 $\pm$ 1.216 |
| GAT | 81.796 $\pm$ 0.661 |
| LRGA + GAT | 85.905 $\pm$ 0.109 |
| GraphSage | 85.039 $\pm$ 0.068 |
| LRGA + GraphSage | 85.229 $\pm$ 0.331 |
| GatedGCN | 85.848 $\pm$ 0.065 |
| LRGA + GatedGCN | 85.944 $\pm$ 0.664 |
| GIN | 85.760 $\pm$ 0.001 |
| LRGA + GIN | **86.765 $\pm$ 0.065** |

## 7 CONCLUSIONS

In this work, we set ourself in a path for improving the generalization power of GNNs. To do so, we introduced the LRGA module, a global self attention module, which is a variant of the dot-product self-attention with linear complexity. In order to theoretically evaluate the contribution of LRGA we analyzed our model under the RGNN framework, which is proved to be universal *in probability*. Under this framework we were able to show that RGNN augmented with LRGA can align with the powerful 2-FWL isomorphism test by learning simple monomial functions, which have a known sample complexity bound. Under certain conditions the latter provides concrete generalization guarantees for RGNN augmented with LRGA. Empirically, we demonstrated augmenting GNN models with LRGA improves their performance significantly, often achieving SOTA performance.

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

## A  PROOF OF PROPOSITION 1

*Proof.* We will now prove the universality *in probability* over the distribution $\mathcal{D}$ of RGNNs. Let $\Omega \subset \mathbb{R}^{n \times d_0} \times \mathbb{R}^{n \times n}$ be a compact set of graphs, $[\boldsymbol{X}, \boldsymbol{A}] \in \Omega$, where $\boldsymbol{X}$ are the node features and $\boldsymbol{A}$ is the adjacency matrix and we assume that $n$ is fixed. Consider $f$, a continuous graph function. $f$ is permutation invariant where the permutation acts on all $n$ dimensions, namely,

$f([\boldsymbol{PX}, \boldsymbol{PAP}^T]) = f([\boldsymbol{X}, \boldsymbol{A}])$ for all permutation matrices $\boldsymbol{P} \in \mathbb{R}^{n \times n}$. RGNN is defined as $\mathrm{RGNN}(\boldsymbol{X}) = \mathrm{GNN}([\boldsymbol{X}, \boldsymbol{R}])$ where $\boldsymbol{R} \in \mathbb{R}^{n \times d}$ are i.i.d. samples from $\mathcal{D}$.

To prove universality in probability we need to show that RGNN can approximate $f$ to an arbitrary precision $\varepsilon$ with high probability $1 - \delta$:

$$\forall \varepsilon, \delta > 0 \quad \exists \Theta, d \ \text{ s.t. } \ P(|\mathrm{RGNN}(\boldsymbol{X}) - f([\boldsymbol{X}, \boldsymbol{A}])| < \varepsilon) > 1 - \delta$$

where $\Theta$ are the RGNN network parameters and $d$ is the dimension of the random features of RGNN.

In fact, a simple RGNN composed of single message passing layer and a global attribute block, a DeepSets network (Zaheer et al., 2017), suffices. The message passing layer first transfers the graph structural information to the node features by creating a factorized representation of $\boldsymbol{A}$. This means that all the graph information is now stored in a set. Then, using the universality of DeepSets network for invariant set functions we can approximate $f$ to an arbitrary precision.

Let us denote the output of the message passing layer of RGNN by $\boldsymbol{h}_1$. The structural information of the graph can be transferred to the node features using the message passing layer by choosing parameters such that $\boldsymbol{h}_1 = [\boldsymbol{X}, \boldsymbol{R}, \boldsymbol{AR}]$. $\boldsymbol{h}_1$ is then fed to the DeepSets network, so we have $\mathrm{RGNN}(\boldsymbol{X}) = \mathrm{DeepSets}([\boldsymbol{X}, \boldsymbol{R}, \boldsymbol{AR}])$.

Observing the approximation error:

$$|\mathrm{RGNN}(\boldsymbol{X}) - f([\boldsymbol{X}, \boldsymbol{A}])| = |\mathrm{DeepSets}([\boldsymbol{X}, \boldsymbol{R}, \boldsymbol{AR}]) - f([\boldsymbol{X}, \boldsymbol{A}])| =$$

$$= |\mathrm{DeepSets}([\boldsymbol{X}, \boldsymbol{R}, \boldsymbol{AR}]) - f([\boldsymbol{X}, \frac{1}{d}\boldsymbol{ARR}^T]) + f([\boldsymbol{X}, \frac{1}{d}\boldsymbol{ARR}^T]) - f([\boldsymbol{X}, \boldsymbol{A}])| \leq$$

$$\leq |\mathrm{DeepSets}([\boldsymbol{X}, \boldsymbol{R}, \boldsymbol{AR}]) - f([\boldsymbol{X}, \frac{1}{d}\boldsymbol{ARR}^T])| + |f([\boldsymbol{X}, \frac{1}{d}\boldsymbol{ARR}^T]) - f([\boldsymbol{X}, \boldsymbol{A}])|$$

We can now bound the two terms in the last inequality above. Since $f$ is defined on the compact set $\Omega$ we first make sure that $\frac{1}{d}\boldsymbol{ARR}^T$ remains bounded (we assume $f$ can be extended continuously to this domain). Since we assume $\mathcal{D}$ is bounded (given $x \sim \mathcal{D}$, $|x| < M/2$), we get:

$$\left\| \frac{1}{d}\boldsymbol{ARR}^T \right\|_F \leq \frac{1}{d} \|A\|_F \|\boldsymbol{RR}^T\| \leq \frac{1}{d} \|A\|_F \, d\frac{M^2}{4} n$$

For the second term we can achieve a bound in probability. Since $f$ is a continuous function on a compact set, by the Heine-Cantor theorem, it is uniformly continuous, meaning that

$$\forall \varepsilon' > 0 \quad \exists \xi \ \text{ s.t } \ \forall \boldsymbol{Q}, \boldsymbol{S} \in \Omega \ \ \mathrm{d}_\Omega(\boldsymbol{Q}, \boldsymbol{S}) < \xi \Rightarrow \ \mathrm{d}_\mathbb{R}(f(\boldsymbol{Q}), f(\boldsymbol{S})) < \varepsilon'$$

Setting $\varepsilon' = \varepsilon/2$ we can now choose $d$ such that with probability $1 - \delta$ we have $\mathrm{d}_\Omega([\boldsymbol{X}, \frac{1}{d}\boldsymbol{ARR}^T], [\boldsymbol{X}, \boldsymbol{A}]) < \xi$. Let $\mathrm{d}_\Omega$ be the euclidean metric, then, $\mathrm{d}_\Omega(\frac{1}{d}\boldsymbol{ARR}^T, \boldsymbol{A}) \leq \|A\|_F \cdot \left\|\frac{1}{d}\boldsymbol{RR}^T - \boldsymbol{I}\right\|_F$. Since we assume a graph of fixed size $n$, $\|A\|_F \leq n$ and we are left with bounding $\left\|\frac{1}{d}\boldsymbol{RR}^T - \boldsymbol{I}\right\|_F$ in probability. Using Hoeffding's inequality we will be able to find $d$ satisfying the conditions.

A single entry in $\boldsymbol{R}$ has mean 0 and variance $c$, for simplicity we set $c = 1$. An entry in $\boldsymbol{RR}^T$ is of the form $(\boldsymbol{RR}^T)_{ij} = \sum_{l=1}^d \boldsymbol{R}_{il}\boldsymbol{R}_{jl}$. Note that all elements of the sum are statistically independent and bounded. Using Hoeffding's inequality:

$$P\left(\left|\frac{1}{d}\sum_{l=1}^d \boldsymbol{R}_{il}\boldsymbol{R}_{jl} - \mathbb{E}\left[\frac{1}{d}\sum_{l=1}^d \boldsymbol{R}_{il}\boldsymbol{R}_{jl}\right]\right| \geq t\right) \leq 2\exp\left(-\frac{2dt^2}{M^4}\right) \tag{8}$$

For $i \neq j$: $\mathbb{E}\left[\frac{1}{d}\sum_{l=1}^d \boldsymbol{R}_{il}\boldsymbol{R}_{jl}\right] = 0$ and for $i = j$: $\mathbb{E}\left[\frac{1}{d}\sum_{l=1}^d \boldsymbol{R}_{il}\boldsymbol{R}_{jl}\right] = 1$.

Using union bound over all entries of $\frac{1}{d}\boldsymbol{RR}^T$:

$$P\left(\bigcup_{i,j \in [n]} \left|\frac{1}{d}(\boldsymbol{RR}^T)_{ij} - \boldsymbol{I}_{ij}\right| \geq t\right) \leq 2n^2 \exp\left(-\frac{2dt^2}{M^4}\right)$$

Setting $t = \xi/n^2 \|\boldsymbol{A}\|_F$ and requiring $2n^2 \exp\left(-\frac{2dt^2}{M^4}\right) < \delta$ we get $d = M' \frac{n^4 \|\boldsymbol{A}\|_F^2}{\xi^2} \log\left(\frac{2n^2}{\delta}\right)$ where $M'$ accumulates all constant factors. Lastly, $\|\boldsymbol{A}\|_F$ is bounded by $n$, so the $d$ we should take is $d = M' \frac{n^6}{\xi^2} \log\left(\frac{2n^2}{\delta}\right)$. Finally, we have that for large enough $d$, $\left\|\frac{1}{d}\boldsymbol{R}\boldsymbol{R}^T - \boldsymbol{I}\right\|_F$ is arbitrarily small with a high probability.

For the first term, we note that $f([\boldsymbol{X}, \frac{1}{d}\boldsymbol{A}\boldsymbol{R}\boldsymbol{R}^T]) = F([\boldsymbol{X}, \boldsymbol{R}, \boldsymbol{A}\boldsymbol{R}])$ is a continuous invariant set function over a bounded domain. Therefore the first term can be bounded by invoking the universal approximation theorem of invariant set functions (Zaheer et al., 2017), i.e., exist a set of parameters and model size such that the approximation error is less than $\varepsilon/2$.

This concludes the proof. We found that exists a set of network parameters and $d$ such that the approximation error is arbitrarily small.

$\square$

## B    MULTISET ENCODING

As shown in Maron et al. (2019a) the multiset encoding function, ENC, can be defined using the collection of Power-sum Multi-symmetric Polynomials (PMPs). That is, given a multiset $\boldsymbol{Z} = (\boldsymbol{z}_1, \ldots, \boldsymbol{z}_n)^T \in \mathbb{R}^{n \times 2d}$ the encoding is defined by

$$\mathrm{ENC}(\boldsymbol{Z}) = \left[\sum_{k=1}^n \boldsymbol{z}_k^{\boldsymbol{\alpha}} \,\middle|\, \boldsymbol{\alpha} \in \mathbb{N}_0^{2d}, |\boldsymbol{\alpha}| \le n\right],$$

where $\boldsymbol{\alpha} = (\alpha_1, \ldots, \alpha_{2d})$, and $\boldsymbol{z}^{\boldsymbol{\alpha}} = z_1^{\alpha_1} \cdots z_{2d}^{\alpha_{2d}}$.

Let us focus on computing a single output coordinate $\boldsymbol{\alpha}$ of the ENC function applied to a particular multiset $\boldsymbol{Z}_{(i,j)}$. This can be efficiently computed using matrix multiplication Maron et al. (2019a): Let $\boldsymbol{\alpha} = (\boldsymbol{\beta}, \boldsymbol{\gamma}) \in \mathbb{N}_0^{2d}$, where $\boldsymbol{\beta}, \boldsymbol{\gamma} \in \mathbb{N}_0^d$. Then,

$$\mathrm{ENC}_{\boldsymbol{\alpha}}(\boldsymbol{Z}_{(i,j)}) = \sum_{k=1}^n \boldsymbol{z}_k^{\boldsymbol{\alpha}} = \sum_{k=1}^n \mathbf{Y}_{i,k}^{\boldsymbol{\beta}} \mathbf{Y}_{k,j}^{\boldsymbol{\gamma}} = (\mathbf{Y}^{\boldsymbol{\beta}} \mathbf{Y}^{\boldsymbol{\gamma}})_{i,j}.$$

By $\mathbf{Y}^{\boldsymbol{\beta}}$ we mean that we apply the multi-power $\boldsymbol{\beta}$ to the feature dimension, i.e., $(\mathbf{Y}^{\boldsymbol{\beta}})_{i,j} = \mathbf{Y}_{i,j}^{\boldsymbol{\beta}}$. This implies that computing the multisets encoding amounts to calculating monomials $\mathbf{Y}^{\boldsymbol{\beta}}, \mathbf{Y}^{\boldsymbol{\gamma}}$ and their matrix multiplications $\mathbf{Y}^{\boldsymbol{\beta}} \mathbf{Y}^{\boldsymbol{\gamma}}$. Thus the 2-FWL update rule, equation 4, can be written in the following matrix form, where for notational simplicity we denote $\mathbf{Y} = \mathbf{Y}^l$:

$$\mathbf{Y}^{l+1} = \left[\left[\mathbf{Y}, \left[\mathbf{Y}^{\boldsymbol{\beta}} \mathbf{Y}^{\boldsymbol{\gamma}} \,\middle|\, (\boldsymbol{\beta}, \boldsymbol{\gamma}) \in \mathbb{N}_0^{2d}, |\boldsymbol{\beta}| + |\boldsymbol{\gamma}| \le n\right]\right]\right]$$

## C    2-FWL VIA POLYNOMIAL KERNELS

In this section, we give a full characterization of feature maps, $\varphi_{\boldsymbol{\beta}}$, of the final polynomial kernel we use to formulate the 2-FWL algorithm. A key tool for the derivation of the final feature map is the multinomial theorem, which we state here in a slightly different form to fit our setting.

**Multinomial theorem.**  Let us define a set of $m$ variables $x_1 y_1, \ldots, x_m y_m$ composed of products of corresponding $x$ and $y$'s. Then,

$$(x_1 y_1 + \cdots + x_m y_m)^n = \sum_{|\boldsymbol{\nu}|=n} \binom{n}{\boldsymbol{\nu}} \prod_{i=1}^m (x_i y_i)^{\nu_i}$$

where $\boldsymbol{\nu} \in \mathbb{N}_0^m$, and the notation $\binom{n}{\boldsymbol{\nu}} = \frac{n!}{\nu_1! \cdots \nu_m!}$. The sum is over all possible $\boldsymbol{\nu}$ which sum to $n$, in total $\binom{n+m-1}{m-1}$ elements.

Recall that we wish to compute $\mathbf{Y}_{i,j}^{\boldsymbol{\beta}}$ as in equation 6 in the paper:

$$\mathbf{Y}_{i,j}^{\boldsymbol{\beta}} = \prod_{l=1}^{d} \left\langle \boldsymbol{x}_i^{s_l}, \boldsymbol{x}_j^{t_l} \right\rangle^{\beta_l} = \prod_{l=1}^{d} \left\langle \varphi_{\beta_l}(\boldsymbol{x}_i^{s_l}), \varphi_{\beta_l}(\boldsymbol{x}_j^{t_l}) \right\rangle = \prod_{l=1}^{d} \left\langle \varphi_{\beta_l}(\boldsymbol{x}_i^s), \varphi_{\beta_l}(\boldsymbol{x}_j^t) \right\rangle$$
$$= \left\langle \varphi_{\boldsymbol{\beta}}(\boldsymbol{x}_i^s), \varphi_{\boldsymbol{\beta}}(\boldsymbol{x}_j^t) \right\rangle$$

We will now follow the equalities in equation 6 to derive the final feature map. The second equality is using the feature maps $\varphi_{\beta_k}$ of the (homogeneous) polynomial kernels (Vapnik, 1998), $\langle \boldsymbol{x}_1, \boldsymbol{x}_2 \rangle^{\beta_k}$, which can be derived from the multinomial theorem.

Suppose the dimensions of $\boldsymbol{X}^{s_l}, \boldsymbol{X}^{t_l}$ are $n \times D_l$ where $\sum_{l=1}^{d} 2D_l = D$. Then, $\varphi_{\beta_l}$ consists of monomials of degree $\beta_l$ of the form $\varphi_{\beta_l}(\boldsymbol{x})_{\boldsymbol{\nu}} = \sqrt{\binom{\beta_l}{\boldsymbol{\nu}}} \prod_{i=1}^{D_l} x_i^{\nu_i} = \sqrt{\binom{\beta_l}{\boldsymbol{\nu}}} \boldsymbol{x}^{\boldsymbol{\nu}}$, $|\boldsymbol{\nu}| = \beta_l$. In total the size of the feaure map $\varphi_{\beta_l}$ is $\binom{\beta_l + D_l - 1}{D_l - 1}$.

The third equality is reformulating the feature maps $\varphi_{\beta_l}$ on the vectors $\boldsymbol{x}_i^s = [\boldsymbol{x}_i^{s_1}, \ldots, \boldsymbol{x}_i^{s_k}] \in \mathbb{R}^{D/2}$, and $\boldsymbol{x}_i^t = [\boldsymbol{x}_i^{t_1}, \ldots, \boldsymbol{x}_i^{t_k}] \in \mathbb{R}^{D/2}$.

The last equality is due to the closure of kernels to multiplication. The final feature map, which is the product kernel, is composed of all possible products of elements of the feature maps, i.e.,

$$\varphi_{\boldsymbol{\beta}}(\boldsymbol{x}) = \left( \prod_{l=1}^{d} \sqrt{\binom{\beta_l}{\boldsymbol{\nu}_l}} \boldsymbol{x}_l^{\boldsymbol{\nu}_l} \ \Big| \ |\boldsymbol{\nu}_j| = \beta_j, \ \forall j \in [d] \right),$$

where $\boldsymbol{x} = [\boldsymbol{x}_1, \boldsymbol{x}_2, \ldots, \boldsymbol{x}_k] \in \mathbb{R}^{D/2}$, and $\boldsymbol{x}_l \in \mathbb{R}^{D_l}$ for all $l \in [d]$. The size of the final feature map is $\prod_{l=1}^{d} \binom{\beta_l + D_l - 1}{D_l - 1} \leq N$ where $N = \binom{n+D}{D}$.

## D  EXTENSION OF PROPOSITION 4

In this section we would like to extend the proof of proposition 4 to the case where the graph is equipped with prior node features $\boldsymbol{X} \in \mathbb{R}^{n \times d_0}$, s.t the network's input is $[\boldsymbol{X}, \boldsymbol{R}]$. As mentioned in Section 3 the isomorphism type of a graph equipped with node features is $\mathbf{Y} = [\![\boldsymbol{I}, \boldsymbol{1} \otimes \boldsymbol{X}, \boldsymbol{X} \otimes \boldsymbol{1}, \boldsymbol{A}]\!]$. Following this description we claim that the node factorization representation of the graph will be of the form $\boldsymbol{R}' = [\boldsymbol{1}, \boldsymbol{X}, \boldsymbol{R}, \boldsymbol{A}\boldsymbol{R}]$, where $\boldsymbol{1} = (1, 1, \ldots, 1)^T \in \mathbb{R}^n$. To build the isomorphosm tensor we can use the sequence of outer products $[\![\boldsymbol{X}_1 \boldsymbol{1}^T, \ldots, \boldsymbol{X}_d \boldsymbol{1}^T, \boldsymbol{1} \boldsymbol{X}_1^T, \ldots, \boldsymbol{1} \boldsymbol{X}_d^T]\!]$, where $\boldsymbol{X}_l \in \mathbb{R}^n$ is the $l$-th column of $\boldsymbol{X}$. This sequence could be represented using the two first components of $\boldsymbol{R}'$. The last two components, $\boldsymbol{R}$ and $\boldsymbol{A}\boldsymbol{R}$ allow to approximate in probability $\boldsymbol{A}$ and $\boldsymbol{I}$ as shown in Appendix A, which complete the isomorphism tensor construction and conclude that $[\boldsymbol{1}, \boldsymbol{X}, \boldsymbol{R}, \boldsymbol{A}\boldsymbol{R}]$ is a node factorization representation. Lastly, we have to show that we can construct this structure using RGNN, and actually we are left to explain how to add the $\boldsymbol{1}$ vector to the representation. This could be done using a global attribute block as used to proof Proposition 1.

## E  SAMPLE COMPLEXITY BOUND OF MONOMIALS

Corollary 6.2 in (Arora et al., 2019) provides a bound on the sample complexity, denoted $\mathcal{C}_{\mathcal{A}'}(g, \epsilon, \delta)$, of a polynomial $g : \mathbb{R}^D \to \mathbb{R}$ of the form

$$g(\boldsymbol{x}) = \sum_j a_j \langle \boldsymbol{\beta}_j, \boldsymbol{x} \rangle^{p_j}, \tag{9}$$

where $p_j \in \{1, 2, 4, 6, 8, \ldots\}$, $a_j \in \mathbb{R}$, $\boldsymbol{\beta}_j \in \mathbb{R}^D$; $\epsilon, \delta$ are the relevant PAC learning constants, and $\mathcal{A}'$ represents an over-parameterized, randomly initialized two-layer MLP trained with gradient descent.

$$\mathcal{C}_{\mathcal{A}'}(g, \epsilon, \delta) = \mathcal{O}\left( \frac{\sum_j p_j |a_j| \|\boldsymbol{\beta}_j\|_2^{p_j} + \log(1/\delta)}{\epsilon^2} \right)$$

It is not immediately clear, however, how to use this theorem to learn an arbitrary monomial $\boldsymbol{x}^{\boldsymbol{\delta}}$ since $g$ has the above particular form. Nevertheless we show how it can be generalized to this case.

Let $\mathcal{B} = \left\{\boldsymbol{\beta} \in \mathbb{N}_0^D \mid |\boldsymbol{\beta}| \le n\right\}$, and note that there are $N = \binom{n+D}{D}$ elements in $\mathcal{B}$. We assume some fixed ordering in $\mathcal{B}$ is prescribed. Define the sample matrix (multivariate Vandemonde) $\boldsymbol{V} \in \mathbb{R}^{N \times N}$ by $\boldsymbol{V}_{\boldsymbol{\alpha}, \boldsymbol{\beta}} = \boldsymbol{\beta}^{\boldsymbol{\alpha}}$. Lemma 2.8 in (Wendland, 2004) implies that $\boldsymbol{V}$ is non-singular. Let $c_{n,D} = \left\|\boldsymbol{V}^{-1}\right\|_\infty$ (i.e., the induced $\ell_\infty$ matrix norm); note that $c_{n,D}$ is dependant only upon $n, D$.

**Lemma 1.** *Fix $D, n \in \mathbb{N}$, and let $\boldsymbol{\delta} \in \mathcal{B}$ be arbitrary. Then, there exist coefficients $\boldsymbol{a} \in \mathbb{R}^N$, $\|\boldsymbol{a}\|_1 \le c_{n,D}$, so that $\boldsymbol{x}^{\boldsymbol{\delta}} = \sum_{\boldsymbol{\beta} \in \mathcal{B}} a_{\boldsymbol{\beta}}(\langle \boldsymbol{\beta}, \boldsymbol{x} \rangle + 1)^n$, for all $\boldsymbol{x} \in \mathbb{R}^D$.*

*Proof.* Using the multinomial theorem we have: $(\langle \boldsymbol{\beta}, \boldsymbol{x} \rangle + 1)^n = \sum_{\boldsymbol{\alpha} \in \mathcal{B}} d_{\boldsymbol{\alpha}} \boldsymbol{\beta}^{\boldsymbol{\alpha}} \boldsymbol{x}^{\boldsymbol{\alpha}}$, where $d_{\boldsymbol{\alpha}}$ are positive multinomial coefficients. This equation defines a linear relation between the monomial basis $\boldsymbol{x}^{\boldsymbol{\delta}}$ and $(\langle \boldsymbol{\beta}, \boldsymbol{x} \rangle + 1)^n$, for $\boldsymbol{\beta} \in \mathcal{B}$. The matrix of this system is $\boldsymbol{V}$ multiplied by a positive diagonal matrix with $d_{\boldsymbol{\alpha}}$ on its diagonal. By inverting this matrix and solving this system for $\boldsymbol{x}^{\boldsymbol{\delta}}$ the lemma is proved. $\square$

We can use this Lemma in the following way: Assume $n$ is even or otherwise consider $2\lceil n/2 \rceil$. Further assume that the MLP $m : \mathbb{R}^{D+1} \to \mathbb{R}$ is two-layer, over-parameterized of the form $m(\boldsymbol{x}, 1)$ (i.e., we assume there is a constant 1 plugged in an extra $D+1$ coordinate). We consider training $m$ with random initialization and gradient descent using data $(\boldsymbol{x}, \boldsymbol{x}^{\boldsymbol{\delta}}) \in \mathbb{R}^D \times \mathbb{R}$ where $\boldsymbol{x}$ is sampled i.i.d. from some distribution $\mathcal{D}$ over $\mathbb{R}^D$.

Let $g : \mathbb{R}^{D+1} \to \mathbb{R}$ defined as $g(\boldsymbol{x}, x_{D+1}) = \sum_{\boldsymbol{\beta} \in B} a_{\boldsymbol{\beta}} \left(\langle \boldsymbol{\beta}, \boldsymbol{x} \rangle + x_{D+1}\right)^n$, where $\boldsymbol{a} \in \mathbb{R}^N$ is as promised by Lemma 1. Then, the learning setup described above is equivalent to training the MLP $m(\boldsymbol{x}, x_{D+1})$ using data of the form $((\boldsymbol{x}, 1), g(\boldsymbol{x}, 1) = \boldsymbol{x}^{\boldsymbol{\delta}})$, where $(\boldsymbol{x}, 1)$ is sampled i.i.d. from a distribution $\mathcal{D}'$ over $\mathbb{R}^{D+1}$ concentrated on the hyperplane $x_{D+1} = 1$. Now using the Corollary 6.2 from (Arora et al., 2019) in our case where $g : \mathbb{R}^{D+1} \to \mathbb{R}$ is defined as $g(\boldsymbol{x}, x_{D+1}) = \sum_{\boldsymbol{\beta} \in \mathcal{B}} a_{\boldsymbol{\beta}} \left(\langle \boldsymbol{\beta}, \boldsymbol{x} \rangle + x_{D+1}\right)^n$ where $\mathcal{B} = \left\{\boldsymbol{\beta} \in \mathbb{N}_0^D \mid |\boldsymbol{\beta}| \le n\right\}$ and by Lemma 1 there exist $\boldsymbol{a}$ such that $g(\boldsymbol{x}, 1) = \boldsymbol{x}^{\boldsymbol{\delta}}$. The sample complexity bound expression by Corollary 6.2 is therefore:

$$\mathcal{C}_{\mathcal{A}'}(g, \epsilon, \delta) = \mathcal{O}\left(\frac{\sum_{\boldsymbol{\beta} \in \mathcal{B}} n\, |a_{\boldsymbol{\beta}}| \left\|\hat{\boldsymbol{\beta}}\right\|_2^n + \log(1/\delta)}{\epsilon^2}\right), \quad \hat{\boldsymbol{\beta}} = (\boldsymbol{\beta}, 1)$$

Let us bound the first term in the numerator of the sample complexity expression:

$$\sum_{\boldsymbol{\beta} \in \mathcal{B}} n\, |a_{\boldsymbol{\beta}}| \left\|\hat{\boldsymbol{\beta}}\right\|_2^n = n \cdot \sum_{\boldsymbol{\beta} \in \mathcal{B}} |a_{\boldsymbol{\beta}}| \left(\sum_{i=1}^D \beta_i^2 + 1\right)^{n/2} \le n \cdot \left(n^2 + 1\right)^{n/2} \sum_{\boldsymbol{\beta} \in \mathcal{B}} |a_{\boldsymbol{\beta}}| \le \left(n^2 + 1\right)^{(n+1)/2} c_{n,D}$$

The first inequality is due to $\|\cdot\|_2 \le \|\cdot\|_1$, the second is by Lemma 1 and uniting $n$ into the main term. From the above, the bound follows.

## F    KERNEL DEFINITION

Let $x, y \in \mathbb{R}^d$ the kernel function were defined in the following manner -

(i) **Polynomial Kernel** - $k_m(x, y) = (\langle x, y \rangle + 1)^m$

(ii) **Exponential Kernel** - $k(x, y) = \exp(\frac{\langle x, y \rangle}{\sqrt{d}})$

(iii) **RBF Kernel** - $k(x, y) = \exp(-\frac{\|x - y\|^2}{\sqrt{d}})$

Lets $\boldsymbol{X}_Q, \boldsymbol{X}_K, \in \mathbb{R}^{n \times d}$ where $\boldsymbol{X}_Q = \{(x_Q^1)^T, \ldots, (x_Q^n)^T\}$ and $\boldsymbol{X}_V = \{(x_V^1)^T, \ldots, (x_V^n)^T\}$ denotes the attention Query and Key matrices. For a given kernel function $k$ we define the attention matrix $\boldsymbol{S} \in \mathbb{R}^{n \times n}$ in the following way -

$$\boldsymbol{S}_{i,j} = \frac{k(x_Q^i, x_K^j)}{\sum_{l=1}^{n} k(x_Q^i, x_K^l)}$$

## G    RANK ABLATION STUDY

We investigated the affects of the attention's rank $\kappa$ on the performance of GNNs augmented with LRGA on the CLUSTER dataset. The dataset contains graphs of 40 to 190 nodes (117 nodes in average). Our experimental setting included fixing the GNN's hidden dimensions size and changing $\kappa$. Figure 1 shows that accuracy increases with the rank values until it reaches a plateau around $\kappa \approx 30$ ($\kappa / \overline{n} = 0.25$ where $\overline{n}$ is the average graph size), a fact that could be attributed to saturating the expressiveness of the LRGA module. Moreover, the maximal accuracy is achieved at a value that corresponds to the maximal graph size in the dataset, smaller than what the theory predicts as a function of the graph size $n$. This rank value is enough to compute any attention function on this graph collection.

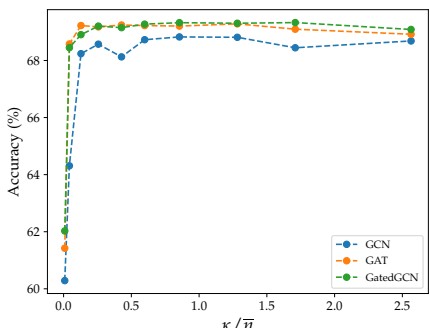

Figure 1: Ablation study on CLUSTER dataset. The X-axis represent the ratio between the rank parameter $\kappa$ and the average graph size $\overline{n} = 117$. The Y-axis represent the network's accuracy

## H    IMPLEMENTATION DETAILS

In this section we describe the datasets on which we performed our evaluation. In addition, we specify the hyperparameters for the experiments section in the paper. The rest of the model configurations are determined directly by the evaluation protocols defined by the benchmarks. It is worth noting that most of our experiments ran on a single Tesla V-100 GPU, if not stated otherwise. We performed our parameter search only on $\kappa$ and $d$ (except for CIFAR10 and MNIST were we searched over different dropout values), since the rest of the parameters were dictated by the evaluation protocol. The models sizes were restricted by the allowed parameter budget.

### H.1    BENCHMARKING GRAPH NEURAL NETWORKS (DWIVEDI ET AL., 2020)

**Datasets.**    This benchmark contains 6 main datasets :

(i) **ZINC**, a molecular graphs dataset with a graph regression task where each node represents an atom and each edge represents a bond. The regression target is a property known as the constrained solubility (with mean absolute error as evaluation metric). Additionally, the node features represent the atom's type (28 types) and the edge features represents the type of connection (4 types). The result reported for GCN used $d = 60$ for the 100K budget and $d = 90$

(network's depth is $L = 12$) for the 500K budget. For the GAT network we used $d = 60$ (4 attention heads of dimension 15) for the 100K budget and $d = 120$ (4 attention heads of dimension 30) with $L = 8$ for the 500K budget. For the GatedGCN network we used $d = 45$ for the 100K budget and $d = 60$ with $L = 12$ for the 500K budget. All the models used $\kappa = 30$.

(ii) **MNIST** and **CIFAR10**, the known image classification problem is converted to a graph classification task using Super-pixel representation (Knyazev et al., 2019), which represents small regions of homogeneous intensity as nodes. The edges in the graph are obtained by applying k-nearest neighbor algorithm on the nodes coordinates. Node features are a concatenation of the Super-pixel intensity (RGB for CIFAR10 and greyscale for MNIST) and its image coordinate. Edges features are the k-nearest distances. The result reported for GCN used $d = 60$ for the 100K budget and $d = 110$ with $L = 8$ for the 500K budget. For the GAT network we used $d = 60$ (4 attention heads of dimension 15) for the 100K budget and $d = 122$ (4 attention heads of dimension 28) with $L = 8$ for the 500K budget. For the GatedGCN network we used $d = 45$ for the 100K budget and $d = 80$ with $L = 8$ for the 500K budget. All the models used $\kappa = 30$.

(iii) **CLUSTER** and **PATTERN**, node classification tasks which aim to identify embedded node structures in stochastic block model graphs (Abbe, 2017). The goal of the task is to assign each node to the stochastic block it was originated from, while the structure of the graph is governed by two probabilities that define the inner-structure and cross-structure edges. A single representative from each block is assigned with an initial feature that indicates its block while the rest of the nodes have no features (CLUSTER), while in the PATTERN dataset nodes are assigned with a random value as input feature at the creation stage. The result reported for GCN used $d = 60$ for the 100K budget and $d = 100$ with $L = 8$ for the 500K budget (PATTERN, CLUSTER respectively). For the GAT network we used $d = 60$ (4 attention heads of dimension 15) for the 100K budget and $d = 120, 60$ (8 attention heads of dimension 15, 4 attention heads of dimension 15) with $L = 8, 12$ for the 500K budget (PATTERN, CLUSTER respectively). For the GatedGCN network we used $d = 45$ for the 100K budget and $d = 80, 50$ with $L = 8, 12$ for the 500K budget (PATTERN, CLUSTER respectively). All the models used $\kappa = 30$.

(iv) **TSP**, a link prediction task that tries to tackle the NP-hard classical Traveling Salesman Problem (Joshi et al., 2019). Given a 2D Euclidean graph the goal is to choose the edges that participate in the minimal edge weight tour of the graph. The evaluation metric for the task is F1 score for the positive class. The result reported for GCN used $d = 60$. For the GAT network we used $d = 60$ (4 attention heads of dimension 15). For the GatedGCN network we used $d = 45$. All the models used $\kappa = 30$.

## H.2 LINK PREDICTION DATASETS FROM THE OGB BENCHMARK (HU ET AL., 2020)

**Datasets.** In order to provide a more complete evaluation of our model we also evaluate it on semi-supervised learning tasks of link prediction. We searched over the same hyperparameter range $\kappa \in \{25, 50, 100\}$, $d \in \{150, 256\}$ and used $\kappa = 50, d = 256$ in all tasks. The three datasets were:

(i) **ogbl-ppa**, an undirected unweighted graph. Nodes represent types of proteins and the edges signify biological connections between proteins. The initial node feature is a 58-dimensional one-hot-vector that indicates the origin specie of the protein. The learning task is to predict new connections between nodes. The train/validation/test split sizes are 21M/6M/3M. The evaluation metric is called Hits@K (Hu et al., 2020).

(ii) **ogbl-collab**, is a graph that represents a network of collaborations between authors. Every author in the network is represented by a node and each collaboration is assigned with an edge. Initial node features are obtained by combining word embeddings of papers by that author (128-dimensional vector). Additionally, each collaboration is described by the year of collaboration and the number of collaborations in that year as a weight. The train/validation/test split sizes are 1.1M/60K/46K. Similarly to the previous dataset, the evaluation metric is Hits@K.

(iii) **ogbl-ddi** - an undirected unwighted graph which represent drug-drug interaction. Each Node represents FDA approved or experimental drug. The edges represent interactions between drugs and represent the joint effect of taking both drugs together. The learning task is to predict new drug to drug interactions. The train/validation/test split sizes are 1M/150K/150K. The evaluation here is also Hits@K.

