# OpenReview forum: "Global Attention Improves Graph Networks Generalization"
_ICLR.cc/2021/Conference — Reject_

### Official Review · AnonReviewer1 · 2020-10-27
**The work lacks novelty**

**Rating:** 5
**Confidence:** 5

**Review:**

The paper proposes the Low-Rank Global Attention (LRGA) module augmented to GNNs to improve generalization power. In particular, given an input graph, the model runs the LRGA module and the GNN module to aggregate node representations on this graph. The input and the outputs of these two modules are concatenated at each layer, followed by a single fully-connect layer (m5) to produce input for the next layer. The LRGA module applies the self-attention mechanism [1], but replacing the softmax layer by the global normalization.

Pros. The results are promising.

Cons.

i) The motivation to propose LRGA by replacing the softmax layer in the self-attention mechanism [1] by the global normalization is not well enough. The graph self-attention networks (such as [3,4]) show competitive results, and they can be applied for large graphs. Thus, LRGA is incremental and not technically sound.

ii) The paper does not discuss the most closely related work, Dual Graph Convolutional Networks (DualGCN) [2]. The architecture LRGA+GNN is similar to DualGCN. Changing from using GCN to another GNN is straightforward, thus the work lacks novelty.

iii) The roles of m1, m2, and m3 are similar to the query, key, and value matrices in the self-attention mechanism, respectively. But why LRGA employs m4? m4 does not have a specific role as it can be placed outside LRGA and put inside Equation 1.
Note that [5] shows that using the vector concatenation/sum-pooling/LSTM over different layers can improve the performance. But, I do not see the role of X^l in Equation 1. What is it?

iv) Given the same GNN module with the same hidden size, the proposed LRGA+GNN has much larger parameters than GNN. This limitation restricts to use of deeper layers.

Minor things: Parentheses in Equation 2 should use between m1 and m2, not m2 and m3.

[1] Attention is all you need. NIPS 2017.
[2] Dual Graph Convolutional Networks for Graph-Based Semi-Supervised Classification. WWW 2018.
[3] Graph-Bert: Only Attention is Needed for Learning Graph Representations. https://arxiv.org/abs/2001.05140
[4] Hyper-SAGNN: a self-attention based graph neural network for hypergraph. ICLR 2020.
[5] Representation Learning on Graphs with Jumping Knowledge Networks. ICML 2018.

=======================
After reading the authors' response:

i. As shown in (new) Table 3 in the revised version, the results of using the global normalization are not better than that of using the softmax layer in the self-attention mechanism. Hence the motivation is not enough.

ii. To have the faster computation, we have SGC[1], FastGCN[2]. To have powerful GNNs, we have GIN[3]. Inspired by DualGCN, we can build a new combination (e.g., SGC+GIN) together with using the vector concatenation/sum-pooling/LSTM over different layers [5] to further improve the performance and have a faster computation. That's reason why the novelty of LRGA+GNN is weak.

[1] Simplifying Graph Convolutional Networks. ICML 2019. [2] Fastgcn: Fast learning with graph convolutional networks via importance sampling. ICML 2018. [3] How Powerful are Graph Neural Networks? ICLR 2019. [5] Representation Learning on Graphs with Jumping Knowledge Networks. ICML 2018.

I keep my score unchanged.

---

> ### Author Response · Authors · 2020-11-19
> **Authors response to reviewer 1**
>
> We thank the reviewer for the comments. We would like to address the raised issues:
>
> **Q -  The motivation to propose LRGA by replacing the softmax layer in the self-attention mechanism [1] by the global normalization is not well enough. The graph self-attention networks (such as [3,4]) show competitive results, and they can be applied for large graphs. Thus, LRGA is incremental and not technically sound.**
>
> **A** -  We would like to highlight several key differences between LRGA and previous attention models:
> First, we performed an experiment comparing LRGA with softmax attention (as well as other common attention kernels) in section 6.3 in the revised paper. Note that LRGA mostly outperforms other attention mechanisms. Furthermore, LRGA acts *globally* on the graph with memory complexity of $O(n\kappa)$ as opposed to [3] which would require quadratic memory complexity with respect to the extracted linkless subgraph size. Regarding [4], Hyper-SAGNN is a model designated to solve the hyperedge prediction problem while our model is designed for classical graph learning tasks. Even though both incorporate a self attention module we don’t feel that the models are very related.
> Second, as far as we know, LRGA is the first global attention mechanism applied to GNNs. Empirically, it improved STOTA in almost all its application to existing GNN models. Theoretically, we analyse the relation of the self attention mechanism augmented with standard GNNs to known graph isomorphism algorithms (2-FWL) and thus justify the benefits gained by using LRGA for graphs.
>
> **Q - Relation to Dual Graph Convolutional Networks (DualGCN) [2].**
>
> **A** -  We respectfully disagree. Despite LRGA+GNN and DualGCN being both composed of two separated modules, a classical GNN and a global aggregator part, there is a major difference between the two models. LRGA is a *learnable* aggregator which allows to aggregate information from different nodes according to the correlation of their node embeddings. On the other hand, the Convp from [2], is a *fixed*  aggregator which is a function of the adjacency matrix, and aggregates information based on the nodes connectivity. Nevertheless, as we agree that diffusion based methods are a related alternative to global attention methods, we added these papers to the related work section in the revised paper.
>
> **Q - The roles of $m_1, m_2$, and $m_3$ are similar to the query, key, and value matrices in the self-attention mechanism, respectively. But why LRGA employs $m_4$? $m_4$ does not have a specific role as it can be placed outside LRGA and put inside Equation 1.**
>
> **A**  -  Thank you for this comment. The observation that m4 could be placed outside of the LRGA module is correct since it does not play a role in the attention mechanism. However, the main reason for including $m_4$ as a part of the LRGA module is to show that the module aligns with the 2-FWL update rule (as described in Equation 7 and the paragraph beneath it). From an empirical point of view, we refer you to section 6.3 in our revised paper, where we added an additional comparison with LRGA without $m_4$, showing inferior performance.
>
> **Q - I do not see the role of $X^l$ in Equation 1. What is it?**
>
> **A** - $X^l$ which can be seen as a skip connection, has a critical role in the main theoretical statements of the the paper:
> - Skip connection allows RGNN to approximate node factorization of the isomorphism type $Y^0$ (proposition 4).
> - From the 2-FWL point of view, the update rule must also include the color encoding of the previous iteration (Equation 5).
> - The first step of the universality proof involves using the skip connection (Proposition 1).
>
> **Q - Given the same GNN module with the same hidden size, the proposed LRGA+GNN has much larger parameters than GNN. This limitation restricts to use of deeper layers.**
>
> **A** -  It is true that augmenting LRGA with GNN without reducing the hidden dimension size leads to a model with a larger number of parameters.  However, as evident in our experiment section, augmenting GNNs with LRGA while keeping the parameter budget *fixed*, outperformed models with larger hidden dimensions. Therefore, the addition of LRGA compensates over the loss of hidden dimension size. The last statement is also demonstrated for deep models in the paper.
>
> **Q - Parentheses in Equation 2 should use between $m_1$ and $m_2$, not $m_2$ and $m_3$.**
>
> **A**  - The parentheses in Equation 2 are located between $m_2$ and $m_3$ in order to exploit the low rank structure of the attention matrix. Placing the parentheses between $m_1$ and $m_2$ enforces the explicit computation of the attention matrix, which corresponds to a quadratic computational cost. Due to the linearity of the normalization factor and the associativity of matrix multiplication replacing the location of the parentheses (to be between $m_2$ and $m_3$) produces an equivalent result while reducing the computational cost to a linear factor.

---

### Official Review · AnonReviewer2 · 2020-10-29
**Good paper. Some improvements possible, but theoretical and empirical results seem solid enough for publication.**

**Rating:** 7
**Confidence:** 4

**Review:**

**Post-discussion update:**

The authors have addressed all my comments. It is unfortunate that a comparison with diffusion graph augmentation could not be added, but I understand the reason provided by the authors and this anyway does not significantly detract from the presented results. Since my original score already marked this as a good paper recommending acceptance, it is left unchanged.

---

**Original review:**

This work proposes to enhance graph neural networks to incorporate global relations between nodes together with local relations learned from the neighborhood structure of the graph. In principle, these global relations are formulated as being computed directly from node features, thus considering the graph as a set of nodes and ignoring the edges between them, but it should be noted that when applied in deeper layers, the node features would encode some edge (or graph) information propagated from the GNN component that still exist in the augmented architecture. To avoid learning or training an unfeasible number of pairwise relations (essentially quadratic in the number of nodes in the graph), the authors propose here to only consider low rank matrices, fixing an upper bound on the rank and thus limiting the learned weights to a reasonable amount. Next, the learned relations between nodes are used together with the ones coming from the graph edges to apply the typical message propagation seen in GNNs (with several such architectures considered here - showing the versatility of the approach as a generally useful module), which establishes the method as implementing an attention mechanism and justifying the name "low rank global attention".

On the theoretical side, the authors establish a relation between their LRGA-augmented approach and the 2-FWL test, which considers coloring of node pairs to produce histograms that encode and compare graph structures. Interestingly, instead of making strict assumptions on the node features provided with the graph, the authors rely here on random features, thus letting the edge composition of the graph dictate the information extracted by the network. Along the way, they also provide interesting insights into the universality of random GNNs, which I must say, may be a bit lost when only presented as a sidenote that is not directly related to the main point of the paper here. Nevertheless, it is an interesting contribution. On the applicative side, the LRGA augmentation is applied to several dominant GNN architectures and generally shows effective benefits in several tasks, benchmarked by carefully following the guidelines provided by recent increasingly-popular attempts (such as the OGB) to standardize comparison protocols in the field of graph representation learning.

The proposed approach here is interesting, well motivated, and supported by solid theoretical and empirical evidence, and therefore I recommend accepting it to the conference. That being said, I have two main suggestions for improvement:

- First, it seems to me that the low rank attention matrix here can also be regarded through the lens of kernel learning. It may be interesting to consider this relation here, and compare (certainly in discussion, or maybe even empirically) the proposed approach to learning attention weights via popular kernel learning methods that also address tractable optimization over what would naively require O(n^2) kernel entries to be learned.

- Second, an alternative approach to attention-based incorporation of less local information, somewhat bypassing edge-based neighborhoods, in GNNs is to a priori modify the structure of the graph to augment it with additional weighted edges. This type of approach can be seen, for example, in "Diffusion Improves Graph Learning" (Klicpera et al., NeurIPS 2019). How does LRGA compare to using such diffusion augmentation? A comparison seems highly warranted here.

Finally, a minor remark: the notations in this paper are a bit difficult to follow when considering concatenations and matrix operations over 3-order tensors. It would be good to properly clarify the relevant notations and operations a bit more rigorously, even though with enough effort one can infer what is going on from context.

---

> ### Author Response · Authors · 2020-11-19
> **Authors response to reviewer 2**
>
> We thank the reviewer for the constructive comments. A revised version of the paper has been uploaded.  We next address the raised issues / refer to changes in the paper:
>
> **Q - First, it seems to me that the low rank attention matrix here can also be regarded through the lens of kernel learning. It may be interesting to consider this relation here, and compare (certainly in discussion, or maybe even empirically) the proposed approach to learning attention weights via popular kernel learning methods that also address tractable optimization over what would naively require O(n^2) kernel entries to be learned.**
>
> **A** - Thank you for this great suggestion: Following this comment we have conducted a series of experiments comparing the performance of LRGA to a collection of known kernel functions (see [1], ref below). The results and relevant discussion can be found in the revised paper in section 6.3.
>
> **Q - Second, an alternative approach to attention-based incorporation of less local information, somewhat bypassing edge-based neighborhoods, in GNNs is to a priori modify the structure of the graph to augment it with additional weighted edges. This type of approach can be seen, for example, in "Diffusion Improves Graph Learning" (Klicpera et al., NeurIPS 2019). How does LRGA compare to using such diffusion augmentation? A comparison seems highly warranted here.**
>
> **A** -  We agree that diffusion based methods are an adequate alternative for exploiting long range dependencies in graphs. Unfortunately, our attempts to incorporate edge diffusion baselines produced inferior performance and we consequently decided not to include it since we feel our implementation might misrepresent the true potential of these methods. Nevertheless, we did add a relevant reference in our related work section.
>
> **Q - The notations in this paper are a bit difficult to follow when considering concatenations and matrix operations over 3-order tensors.**
>
> **A** - We modified and clarified the concatenation notation in Section 3 of the revised version.
>
> [1] Tsai, Y. H. H., Bai, S., Yamada, M., Morency, L. P., & Salakhutdinov, R. (2019). Transformer Dissection: An Unified Understanding for Transformer's Attention via the Lens of Kernel. EMNLP 2019.

---

### Official Review · AnonReviewer3 · 2020-10-29
**Comments**

**Rating:** 6
**Confidence:** 4

**Review:**

This work proposes a Low-Rank Global Attention (LRGA) module for GNN which is more efficient in terms of computation and memory than the dot-product attention. Then the authors prove that LRGA augmented RGNN algorithmically aligns with a single head 2-FWL update step. The authors claim that if one network has algorithmically alignment with an algorithm, then the network has good generalization guarantee. Based on this, the network with the proposed LRGA can enjoy better generalization performance.

To be honest, I am not familiar to GNN and its related theory. For a general reader, I have the following questions.

Metric:
1)	Good theoretical guarantees. The authors well prove the algorithmically alignment of LRGA. To be honest, I am not familiar to GNN and its related theory. So if as claimed by the authors that if one network has algorithmically alignment with an algorithm, the network has good generalization guarantee, then the theoretical results in this work can guarantee better performance of the proposed LRGA.
2)	The experimental results also show the superiority of LRGA. When equipped with LRGA, the GNN baselines can enjoy much better results. But as I mentioned, I am not familiar to GNN, and am not very sure whether the compared methods are state-of-the-arts in this field.

Some issues.
1)	The low-rank based attention is not novel actually. Low-rank based models are well studied in model compression, data structure modeling, in which most of the works also aim to improve the generalizations by reducing redundant parameters. So here the authors use the low-rankness to improve the generalization is not new.
2)	The authors claim that if one network has algorithmically alignment with an algorithm, the network has good generalization guarantee. This is hard to understand intuitively. Since this is the main basic claim in this work, the authors should well explain it.

---

> ### Author Response · Authors · 2020-11-19
> **Authors response to reviewer 3**
>
>
> We thank the reviewer for the constructive comments. A revised version of the paper has been uploaded. We next address the raised issues / refer to changes in the paper:
>
> **Q  - The authors claim that if one network has algorithmically alignment with an algorithm, the network has good generalization guarantee. This is hard to understand intuitively. Since this is the main basic claim in this work, the authors should well explain it.**
>
> **A** - Thank you for this comment: we added an explanation to the revised version in section 5.2 (algorithmic alignment). Note that revisions are colored in blue for easier navigation.
>
> **Q - The low-rank based attention is not novel actually. Low-rank based models are well studied in model compression, data structure modeling, in which most of the works also aim to improve the generalizations by reducing redundant parameters. So here the authors use the low-rankness to improve the generalization is not new.**
>
> **A** -  First, the use of low rank attention in the context of GNNs is new. In fact, as far as we know, we introduce the first *global* attention mechanism on graph data. This is made possible due to the low rankness of the attention mechanism.
> Second, our suggested attention is a modified version of the standard attention, where we add $m_4$ (additional mlp) acting on the nodes separately, which is motivated in our theoretical analysis; please see revised paper section 6.3 (attention ablation) where we demonstrate incorporating $m_4$ has some practical advantages over standard attention mechanisms.
>
> Third, the improved generalization in our case is not the outcome of reducing redundant variables; this is demonstrated empirically in Appendix G, where we show increasing the rank does not impair generalization. The main claim in our paper is that the improved generalization is attributed to the algorithmic alignment with 2-FWL, achieved by augmenting GNNs with LRGA (which we prove in section 5.3).

---

### Official Review · AnonReviewer4 · 2020-10-31
**Official Blind Review #4**

**Rating:** 6
**Confidence:** 3

**Review:**

##########################################################################

Summary

The paper presents the Low-Rank Global Attention module that can be incorporated into Graph Neural Networks (GNN) to improve their generalization power. It analyses and proves the Random Graph Neural Network is universal in probability but with limited generalization, while the RGNN augmented with LRGA aligns with a powerful graph isomorphism test, namely the 2-FolkloreWeisfeiler-Lehman (2-FWL) algorithm. The results validate the effectiveness of augmenting existing GNN layers with LRGA.

##########################################################################

Pros:
+ Overall, the paper is well written and structured. The theoretical analysis of RGNN and its augmentation with LRGA is interesting.
+ Extensive experiments are conducted, and the results show the effectiveness of LRGA.

##########################################################################

Cons:
- The main concern of this paper is its similarity to a missing reference [Puny et al., 2020]. In that paper, the exact same low-rank global attention module is proposed and incorporated into GNN to improve the performance. A similar theoretical analysis is also conducted in [Puny et al., 2020]. A detailed comparison with [Puny et al., 2020] is expected.

- From Table 1, we can see that the argumentation with LRGA has different effects on different GNNs and with different parameter budget. For example, the improvement on GatedGCN is less compared to that on the others. The reasons behind this are expected.

Missing reference:
Puny et al. From Graph Low-Rank Global Attention to 2-FWL Approximation. In Proc. of ICML Workshop Graph Representation Learning and Beyond, 2020.

##########################################################################

Minor issue:
* In Section 6.3, it is claimed that GraphSage achieves an accuracy of 50.49%, and it seems a typo. It should be 81.25%.

##########################################################################
After discussion:
Since the authors and other reviewers have addressed my concerns, I would like to change my score. The introduction of LRGA into the RGNN is interesting with the theoretical analysis, although the network design is incremental. I am now leaning towards borderline acceptance.

---

> ### Author Response · Authors · 2020-11-19
> **Authors response to reviewer 4**
>
> **Q  - The main concern of this paper is its similarity to a missing reference [Puny et al., 2020]**
>
> **A** - We are not able to address the main concern raised in this review due to the ICLR submission guidelines.
>
> **Q - From Table 1, we can see that the argumentation with LRGA has different effects on different GNNs and with different parameter budget. For example, the improvement on GatedGCN is less compared to that on the others. The reasons behind this are expected.**
>
> **A** - One explanation could be that the GatedGCN incorporates a sort of soft attention process on the edges [1]; although local, it already introduces an effective inductive bias allowing for good generalization in some cases. We can incorporate such a discussion in the experiments if the reviewer feels it's helpful.
>
> **Q - In Section 6.3, it is claimed that GraphSage achieves an accuracy of 50.49%, and it seems a typo. It should be 81.25%.**
>
> **A** - The discrepancies in the values is due to the multiple versions of the benchmark. Our reported value (50.49%) is according to the latest version while 81.25% was the reported value in an older version.
>
> [1] Vijay Prakash Dwivedi, Chaitanya K. Joshi, Thomas Laurent, Yoshua Bengio, and Xavier Bresson.Benchmarking graph neural networks, 2020

---

### Author Response · Authors · 2020-11-25
**Thank you for the reviews**

We would like to thank the reviewers for the time they dedicated for evaluating our work. We believe that their
important remarks helped us further improve our paper.

---

### Decision · Program_Chairs · 2021-01-07
**Final Decision**

**Decision:**

Reject

**Comment:**

The paper provides some contribution to the field by exploiting in an unexplored  way background knowledge already covered by relevant literature. Presentation of the proposal is well structured and clear. The paper is also providing interesting theoretical and experimental results. The theoretical results, however, could be better explained.
Overall the proposed work seems to be incremental. Perhaps a deeper investigation into the relationships with diffusion augmentation would add more value to the contribution.